# Ligand dependent interaction between PC-TP and PPARδ mitigates diet-induced hepatic steatosis in male mice

Samuel A. Druzak[1,6], Matteo Tardelli[2,6], Suzanne G. Mays[1], Mireille El Bejjani[1], Xulie Mo[3], Kristal M. Maner-Smith[4], Thomas Bowen[4], Michael L. Cato[1], Matthew C. Tillman[1], Akiko Sugiyama[2,5], Yang Xie[2,5], Haian Fu[3], David E. Cohen[2,5] & Eric A. Ortlund[1] ✉

Phosphatidylcholine transfer protein (PC-TP; synonym StarD2) is a soluble lipid-binding protein that transports phosphatidylcholine (PC) between cellular membranes. To better understand the protective metabolic effects associated with hepatic PC-TP, we generated a hepatocyte-specific PC-TP knockdown (L-$Pctp^{-/-}$) in male mice, which gains less weight and accumulates less liver fat compared to wild-type mice when challenged with a high-fat diet. Hepatic deletion of PC-TP also reduced adipose tissue mass and decreases levels of triglycerides and phospholipids in skeletal muscle, liver and plasma. Gene expression analysis suggest that the observed metabolic changes are related to transcriptional activity of peroxisome proliferative activating receptor (PPAR) family members. An in-cell protein complementation screen between lipid transfer proteins and PPARs uncovered a direct interaction between PC-TP and PPARδ that was not observed for other PPARs. We confirmed the PC-TP− PPARδ interaction in Huh7 hepatocytes, where it was found to repress PPARδ-mediated transactivation. Mutations of PC-TP residues implicated in PC binding and transfer reduce the PC-TP-PPARδ interaction and relieve PC-TP-mediated PPARδ repression. Reduction of exogenously supplied methionine and choline reduces the interaction while serum starvation enhances the interaction in cultured hepatocytes. Together our data points to a ligand sensitive PC-TP− PPARδ interaction that suppresses PPAR activity.

Steroidogenic acute regulatory lipid transport (START/STARd) domain-containing proteins bind and transport glycerophospholipids, ceramides, and sterols. This family is composed of 15 members that are further classified based on sequence and ligand preferences. The STARD2 subfamily, containing STARD2, 7, and 10, bind glycerophospholipids species. The titular member of this family, phosphatidylcholine transfer protein (PC-TP a.k.a STARD2), specifically binds and transfers phosphatidylcholine (PC) generated in the endoplasmic reticulum throughout the cell to modulate plasma membrane, lipid, and thermal homeostasis[1–3]. Knockout of PC-TP ($Pctp^{-/-}$) in mice leads to beneficial alterations in fatty acid metabolism, glucose homeostasis, and liver health. In the context of

[1]Department of Biochemistry, Emory University School of Medicine, 1510 Clifton Road, Atlanta, GA, USA. [2]Joan & Sanford I. Weill Department of Medicine, Weill Cornell Medical College, New York, NY, USA. [3]Department of Chemical Biology and Pharmacology, Emory University School of Medicine, 1510 Clifton Road, Atlanta, GA, USA. [4]Emory Integrated Lipidomics and Metabolomics Core, Emory University School of Medicine, 1510 Clifton Road, Atlanta, GA, USA. [5]Division of Gastroenterology, Hepatology and Endoscopy, Brigham and Women's Hospital, Harvard Medical School, Boston, MA, USA. [6]These authors contributed equally: Samuel A. Druzak, Matteo Tardelli. ✉e-mail: eortlun@emory.edu

high-fat diet, $Pctp^{-/-}$ protects mice from diet-induced insulin and glucose intolerance as indicated by glucose tolerance test and increases in phosphorylation of important mediators of insulin signaling[4]. $Pctp^{-/-}$ mice are also protected from hepatoxicity induced by a methionine and choline-deficient (MCD) diet[5]. Therefore, PC-TP promotes pathological effects in multiple models of liver damage.

Many of the phenotypic effects observed in $Pctp^{-/-}$ mice have been rationalized in the context of its interaction with thioesterase superfamily member 2 (THEM2). In the cytoplasm near the mitochondrial membrane, THEM2 cleaves fatty-acyl-CoA into free fatty acids, which may then be conjugated and imported into the mitochondria by CPT1a– ACSL. The PC-TP– THEM2 interaction increases the thioesterase activity of THEM2, resulting in increased fatty acid metabolism and improved glucose and insulin tolerance[6]. Interestingly, knockout of PC-TP, but not THEM2, improves liver health and increases peroxisome proliferative activating receptor (PPAR) activity, suggesting THEM2 does not solely mediate the effects of PC-TP[7].

PPARs are part of a lipid-sensing nuclear receptor family comprised of three members: PPARα, γ, δ. Members of this family have the largest ligand-binding pocket of any nuclear receptor allowing them to accommodate a variety of ligands from fatty acids to phosphatidylcholines (PCs) and their metabolites[8–12]. PPARs are localized primarily in the nucleus and heterodimerize with retinoid × receptor alpha (RXRα) to bind DNA and control transcription of genes involved in metabolism, proliferation and inflammation. Ligand binding induces shedding of co-repressor complexes by inducing an allosteric shift in the activation function surface 2 (AF-2). This allows for coactivator association, which modulates transcription by recruiting enzymes involved in DNA methylation or histone acetylation[13]. It is unknown how lipophilic ligands are transported to PPARs in the nucleus. One possible mechanism is through delivery by lipid transport proteins (LTPs). For example, fatty-acid-binding proteins (FABPs) have been shown to act as lipid chaperones delivering ligands to the PPAR family in an isoform-specific manner[14–19]. However, FABPs only bind a subset of reported PPAR ligands and it is unclear how PPARs access larger ligands such as glycerophospholipids.

We sought to characterize the role of PC-TP in the liver to determine the mechanism by which hepatic PC-TP modulates energy homeostasis. Transcriptomic analysis of livers isolated from $Pctp^{-/-}$ vs. wild-type mice fed an MCD diet to induce steatosis revealed PC-TP-dependent alterations in PPAR signaling that were enhanced upon methionine and choline restriction. We generated a transducible, liver-specific PC-TP knockout (L-$Pctp^{-/-}$), which had a mild phenotype under normal dietary conditions but reduced hepatic lipid accumulation, improved glucose and insulin homeostasis and increased endurance during exercise when given a high-fat diet. Targeted gene expression analysis showed induction of PPAR-controlled transcripts in L-$Pctp^{-/-}$ mice fed a high-fat diet, suggesting PC-TP represses PPARδ during overnutrition. We demonstrate that PC-TP interacts directly with PPARδ to suppress its transcriptional activity, using a combination of cellular assays and binding assays with purified proteins. The PPARδ ligand-binding domain (LBD) is sufficient for PC-TP and FABP5 interaction in vitro, though full-length PPARδ (FL-PPARδ) is required in cells. Mutations to the PC-TP lipid-binding pocket abrogate PPARδ interaction and relieve the observed gene suppression. Taken together, these observations suggest a role for PC-TP in regulating PPARδ.

## Results

### RNA-seq analysis of liver tissue from $Pctp^{-/-}$ mice
Previous work has shown positive metabolic effects associated with whole-body knockout (KO) of PC-TP in mice ($Pctp^{-/-}$). $Pctp^{-/-}$ mice are more sensitive to glucose and insulin, have increased beta oxidation and improved liver health compared to wild-type mice[3,4,20]. When given a methionine and choline-deficient (MCD) diet to induce liver damage, $Pctp^{-/-}$ mice develop steatosis but are protected from hepatotoxicity[5].

To better understand mechanisms underlying the protective effects associated with PC-TP deletion, we investigated changes in the transcriptome that occur in the livers of $Pctp^{-/-}$ mice on normal chow and on the MCD diet.

$Pctp$ KO changes 97 genes in mice given normal diet (ND) and 415 genes in mice on the MCD diet. Under both dietary conditions, we detected a large subset of genes known to be regulated via PPARs (Fig. 1A, B). This observation was confirmed when cross referencing the differentially expressed genes (DEGs) with the ChIP Enrichment Analysis (CHEA) database (Supplemental Fig. 1A, B)[21,22]. Kyoto Encyclopedia of Genes and Genomes (KEGG) pathway analysis revealed increased perturbations in PPAR-associated processes in the MCD condition compared to ND (Supplemental Fig. 1C, D)[23]. Comparing the effect of diet within each genotype shows a reduction of differential PPAR regulation for both CHEA and KEGG analysis when comparing WT (Fig. 1C, D) to $Pctp^{-/-}$ mice (Fig. 1E, F) supporting a role for PC-TP in differential PPAR regulation resulting from MCD diet.

### In vivo characterization of L-$Pctp^{-/-}$
We established an inducible hepatocyte-specific PC-TP KO (L-$Pctp^{-/-}$) mice to avoid potential confounding developmental or compensatory effects from a whole-body deletion and to query the specific role of hepatic PC-TP. L-$Pctp^{-/-}$ mice were generated through i.v. injection of AAV8-TBG-Cre into $Pctp^{flox/flox}$ mice. AAV8 harboring CRE under the TBG promoter ensures hepatocyte-specific deletion. Mice were challenged with either high-fat diet (HFD) or normal diet (ND) (Fig. 2A, B). When given a ND, L-$Pctp^{-/-}$ mice showed no significant change in body weight, glucose tolerance compared to WT mice (Fig. 2C–E). L-$Pctp^{-/-}$ mice displayed increased gluconeogenesis in the liver (Fig. 2F) and only minor alterations in the composition of liver and serum and liver lipids (Supplemental Fig. 2A–J).

Under HFD feeding, L-$Pctp^{-/-}$ mice gained less weight and had reduced adipose tissue mass relative to their WT counterparts (Fig. 2C). This change in weight started on week 5 and continued throughout the assay period (Supplemental Fig. 3A). Food and water intake between HFD-fed mice were not significantly different for WT or L-$Pctp^{-/-}$ mice, suggesting these effects are independent of nutrient consumption (Supplemental Fig. 3B, C). We next measure glucose levels in response a bolus injection of glucose, insulin, or pyruvate (Fig. 2C–E). Quantifying these data using total area under the curve show improved glycemic control (Fig. 2D) and insulin sensitivity (Fig. 2E). These changes in glucose and insulin homeostasis were also reflected in decreased fasting levels of both glucose and insulin (Supplemental Fig. 3D, E).

In addition to a reduction in fat mass and total body weight, we also observed a significant decrease in liver weight for L-$Pctp^{-/-}$ mice fed HFD compared to WT HFD mice (Fig. 3A). Under HFD, deletion of hepatic PC-TP normalized liver size to a weight comparable to WT mice fed ND. In line with this, we also observed reduced lipid droplet formation in L-$Pctp^{-/-}$ mice fed HFD relative to WT HFD mice (Fig. 3B, C), accompanied by decreases in hepatic triglyceride (TG) content (Fig. 3D, E). We complimented this analysis by performing untargeted lipidomics on livers isolated from each animal (Supplemental Data 1). Similar to previous analysis, only triglyceride species seemed to be significantly changed upon dietary intervention or deletion of PC-TP (Supplemental Fig. 4A, B). Most triglyceride species that were altered contained an essential fatty acid in at least one position. Taken together, this data points to a role for PC-TP in the progression of liver lipotoxicity driven by a state of overnutrition.

We next investigated whether the decreased liver cholesterol and TG could be a result of increased export of these lipid species into the serum. Serum lipid profiling reveals similar levels of TGs, PLs, and nonesterified free fatty acids (NEFA) for all groups. However, there is an increase in total serum cholesterol levels in KO HFD, likely due to changes in the composition of lipoprotein particles (Fig. 3F–J).

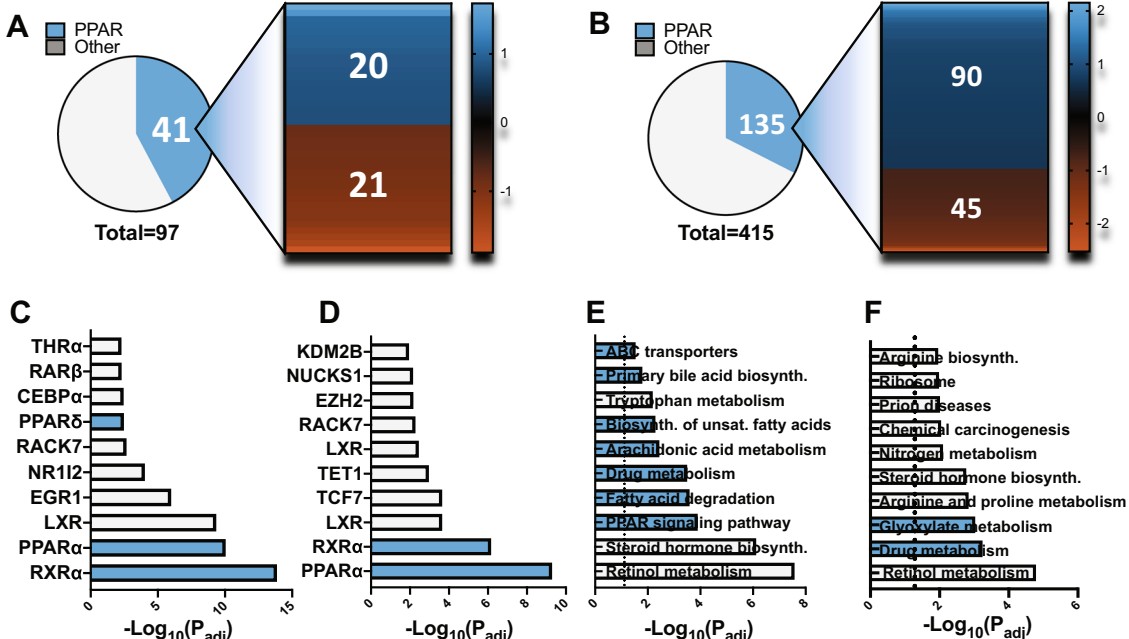

**Fig. 1 | RNAseq comparing *Pctp*⁻/⁻ and WT chow and MCD fed mice.** RNA from chow and MCD fed WT (*n* = 3) and *Pctp* ⁻/⁻ (*n* = 3) mouse livers was used for RNAseq analysis. Statistical significance was determined using the Wald test followed by the Benjamini Hochberg correction. **A**, **B** Differentially expressed genes (DEGs) for chow fed and MCD fed mice, respectively, comparing WT to PC-TP KO. Heat Map shows the distribution of PPARδ controlled DEGs. **C**, **D** Enrichr analysis of DEGs comparing the effect of diet on each genotype compared to transcription factor CHIP-seq databases (CHEA), for WT and *Pctp*−/−, respectively[21,22]. **E & F** Enrichment analysis of altered metabolic pathways (KEGG) determined by cross referencing the statistically significantly altered genes comparing the effect of diet on each genotype, for WT and *Pctp*−/−, respectively. Blue denotes a PPAR related process[22,23]. Source data are provided as a Source Data file.

Supporting this idea, we see more cholesterol packaged into the lipoprotein particles as well as less triglyceride content (Fig. 3K, L).

The HFD L-*Pctp*⁻/⁻ mice do not have significant alterations in fatty acid oxidation, energy expenditure or respiration exchange ratio (Fig. 4A–C). We also interrogated the lipid profile of muscle isolated from HFD mice and found that L-*Pctp*⁻/⁻ HFD mice had decreased levels of TG, NEFA and phospholipid species compared to WT HFD mice (Fig. 4D–F). Taken together these observations point to alterations in lipid homeostasis without affecting substrate utilization, energy expenditure or food intake. HFD L-*Pctp*⁻/⁻ mice displayed an increase in distance traveled during voluntary wheel running when compared to WT mice (Fig. 4G). Follow up endurance running tests showed ND L-*Pctp*⁻/⁻ trended toward running further and longer than control mice (Fig. 4H, I).

Taken together our data suggest a role for PC-TP in regulating PPARs as our L-*Pctp* ⁻/⁻ mouse on HFD displayed an opposing phenotype to hepatic deletion of PPARs. Deletion of hepatic PPARα resulted in decreased beta oxidation, insulin resistance and increased liver lipid accumulation. Similarly liver-specific PPARδ KO presented with insulin resistance, dyslipidemia, and steatosis[24,25]. While seemingly redundant, PPARα is thought to control these processes in the fasted state, with PPARδ playing a more important role in the fed state[26]. To test this, we performed a PPAR qRT-PCR array on livers from HFD animals, which shows an increase in PPAR regulated transcripts in L-*Pctp*⁻/⁻ mice (Fig. 4J) (Supplemental Fig. 3F).

### Defining the PC-TP–PPAR interactome: discovery of a repressive interaction between PC-TP and PPARδ

We systematically interrogated the ability of PPARs to interact with STARD and FABP proteins using a split nano-luciferase protein complementation assay (nano-PCA) (Fig. 5A) developed within the Emory Chemical Biology Discovery Center[27]. This in-cell approach relies on the reversible interaction between two fragments of nano-luciferase whereby two proteins containing either the N- or C-terminal portion of

nano-luciferase interact to permit reconstitution of the intact enzyme. We observed a PC-TP–PPARδ interaction signal that is stronger in this assay than literature-reported interactors for both PC-TP (PAX3, Them2) and PPARδ (FABP5) (Fig. 5B, C)[16,28]. In agreement with in vivo studies, transient knockdown of PC-TP in Huh7 cells leads to an increase in PPARδ activity as measured by luciferase reporter activity (Fig. 5D) implying an inhibitory role for PC-TP in regulating PPARδ activity. Knockdown efficiency was measured via qPCR and Western blot (Supplemental Fig. 5A–C). We verified the repressive effect of PC-TP on PPARδ using qRT-PCR. Consistent with luciferase reporter data, we observed an increase in the PPARδ target genes *ANGPLT4*, *HMGCS2*, *ADRP*, and *MCAD* upon knockdown of PC-TP. Expression of PPARδ was unchanged by the knockdown (Fig. 5E).

### Domain mapping of the PPARδ–PC-TP and PPARδ–FABP5 complex

There are four functionally important surfaces that FABP5 or PC-TP could bind to affect PPARδ transactivation: the DNA-binding domain, N-terminal activation function domain 1 (AF-1), activation function surface (AF-S), or the heterodimer interface (Fig. 6A). The AF-1 and AF-S are implicated in ligand-independent and ligand-dependent activation, respectively, whereas the DNA binding and heterodimer interface are where PPARδ binds DNA or RXRα, respectively. In-cell PCA assays showed that full-length PPARδ generates the strongest interactions with PC-TP and FABP5 compared to any isolated domain, suggesting the interaction between LTP and PPARs is multivalent or cooperative (Fig. 6B). Previous reports suggest that FABPs modulate PPARs in a ligand-dependent manner involving direct ligand transfer[17]. We therefore expected the main interaction interface to be located somewhere within the ligand-binding domain (LBD). Instead, FABP5 interacts more strongly with the PPARδ N-terminal AF-1-DBD than the LBD alone, though still at levels lower than observed for the full-length protein (Fig. 6D). To characterize these interactions in vitro we monitored the ability of purified Cy5-labeled PPARδ LBD or full-length (FL) PPARδ to

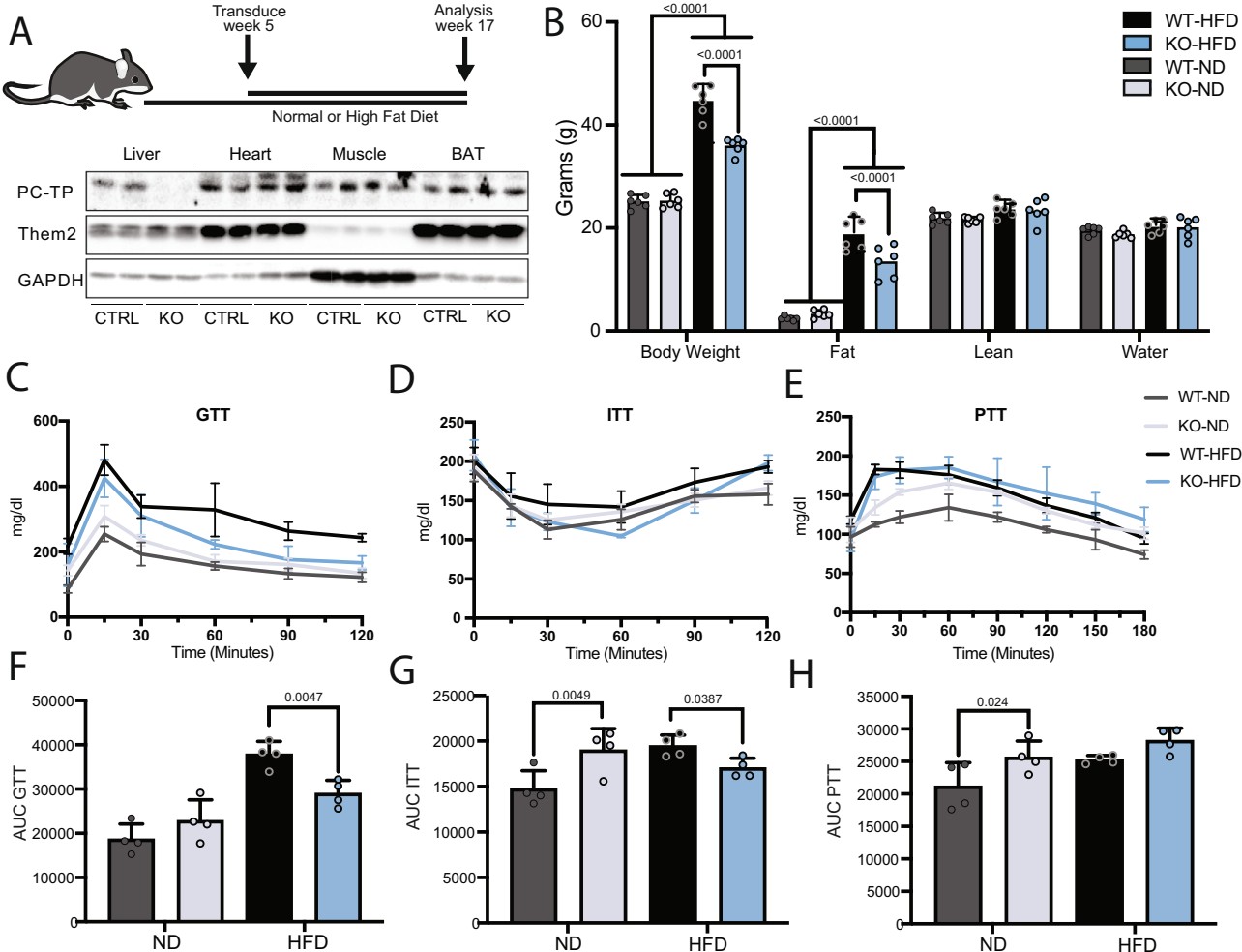

**Fig. 2 | Characterization of the L-*Pctp*⁻/⁻ mouse. A** L-*Pctp*⁻/⁻ mice were generated through i.v. injection of AAV8-TBG-Cre into *Pctp*^flox/flox^ mice. AAV8 harboring CRE under the TBG promoter ensures hepatocyte-specific deletion. Control mice (WT) were *Pctp*^flox/flox^ mice treated with AAV8-TBG-LacZ through i.v. injection. Following AAV8 injection, mice were fed either ND or HFD for an additional 12 weeks prior to analysis. Representative western blot showing hepatocyte-specific knockdown of PC-TP in L-*Pctp*⁻/⁻ group (*n* = 2). **B** Echo-RI of L-*Pctp*⁻/⁻ and WT mice on ND or HFD (*n* = 6, two-way ANOVA, SEM). **C–E** Curves for the GTT, ITT and PTT used for mapping total area under the curves measured after 12 weeks of diet intervention (*n* = 4, SEM). **F** Total area under the curve for glucose tolerance test (*n* = 4, one-way ANOVA, SEM). **G** Total area under the curve for insulin tolerance test (*n* = 4, one-way ANOVA, SEM). **H** Total area under the curve for pyruvate tolerance test (*N* = 4, one-way ANOVA, SEM). Source data are provided as a Source Data file.

bind purified PC-TP or FABP5 using temperature related intensity change (TRIC) on a Dianthus Pico 2.3 (NanoTemper) (Fig. 6C, E)[29]. Both FABP5 and PC-TP interact with FL-PPARδ with binding affinities (Kd) of ~150 nM and ~2.4 μM, respectively. In contrast to in-cell domain mapping, there was a less pronounced difference in binding between PPARδ-LBD or FL-PPARδ with either PC-TP or FABP5.

### Altered lipid levels modulate the interaction between PC-TP or FABP5 with PPARδ

As shown using purified components, the PPARδ-LBD is required for PC-TP interaction, and our in vivo data suggests modulation of hepatic PPARδ could be dependent on lipid availability. This is in line with idea that PC-TP may sense or channel a ligand to PPARδ, and we hypothesized that the interaction could be dependent on PC-TP binding to a lipid ligand. To test this, we either lowered the concentration of exogenously supplied lipids via serum starvation or grew cells in media depleted of methionine and choline and interrogated complex formation by performing Bioluminescence Resonance Energy Transfer (BRET) assays[30]. The PC-TP–PPARδ interaction was increased in the serum restricted condition when compared to cells cultured in full media. This increase in the PC-TP–PPARδ interaction is juxtaposed by a

decrease in the interaction between FABP5 and PPARδ (Supplementary Fig. 5A). In line with our hypothesis that the interaction between PC-TP and PPARδ is modulated by ligand, we see enhanced repression of PPARδ activity when cells are restricted of serum (Supplementary Fig. 5B). While serum starvation increased PC-TP–PPARδ complex formation, culturing cells in MCD media ablated the interaction (Fig. 7A). PC-TP preferentially binds PCs with medium-length, saturated SN1 acyl-chains and long, polyunsaturated SN2 acyl-chains[31,32]. We therefore used targeted lipidomic analysis to profile PC levels in ER from livers isolated from *Pctp*⁻/⁻ fed either HFD or ND, which showed PC-TP-dependent PC changes, in particular PC (36:4) (Fig. 7B) (Supplemental Fig. 6C).

### Ligand binding and transfer enhance PC-TP–PPARδ complex formation

To understand how PC-TP interacts with its preferred ligand, we solved the structure of PC-TP bound to PC (16:0/20:4) to 2.18 Å, which contains an arachidonoyl acyl chain in the SN2 position (PDB:7U9D) (Supplemental Table 1). PC-TP adopted the expected STARd fold consisting of nine antiparallel beta sheets with four alpha helices. Our structure revealed clear electron density to guide the modeling of the bound ligand (Fig. 7C, D). The PC (16:0/20:4) phosphate group is

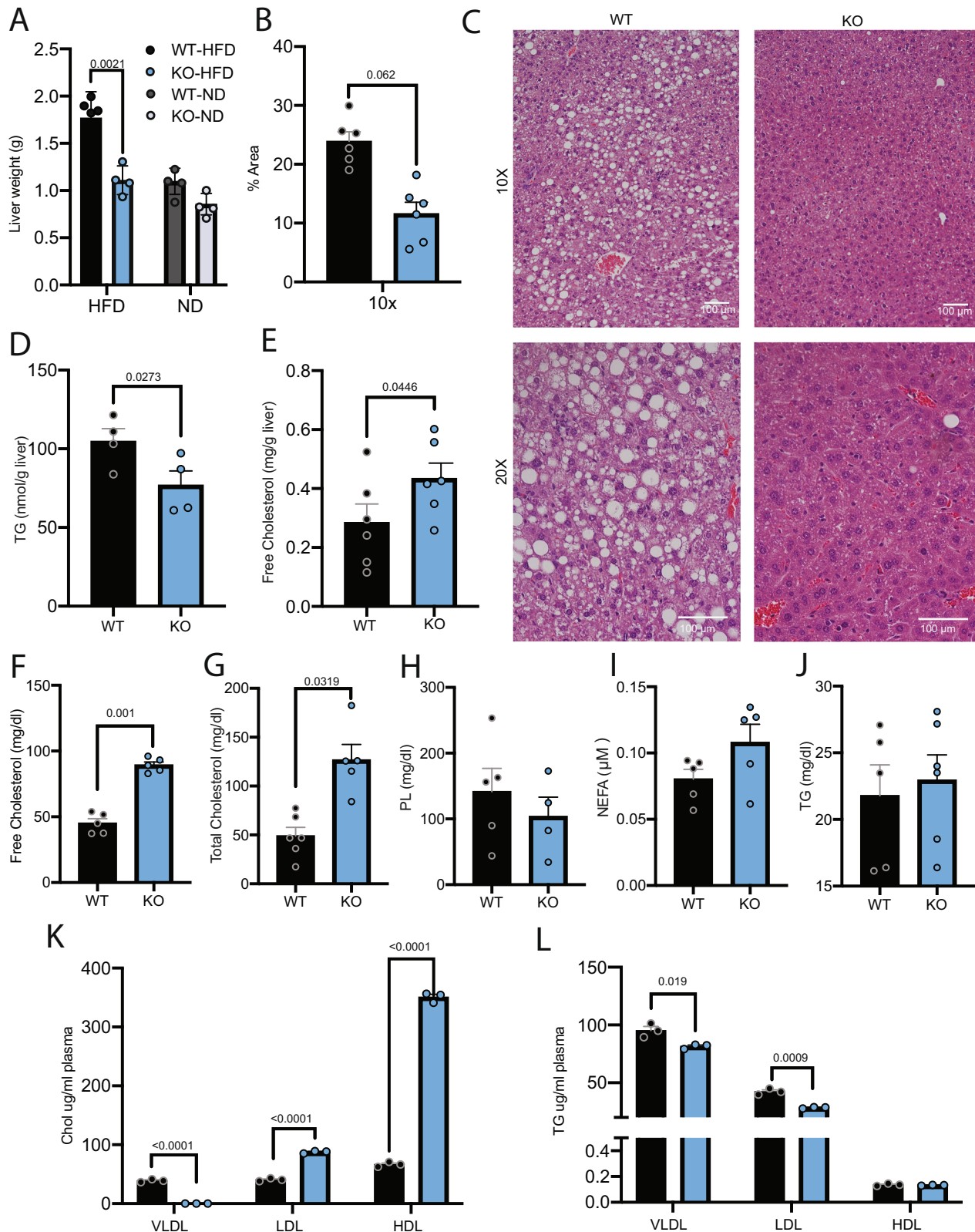

coordinated by Tyr72, Arg78, and Gln157. Whereas the quaternary ammonium on the choline head group is stabilized by an aromatic cage consisting of Trp101, Tyr114, Tyr116, and Tyr155 (Fig. 7E). Using this structure, we designed mutations to reduce phospholipid binding (R78E, R78A, and Y114R). Mutating Arg78 to glutamic acid (R78E) introduces a charge repulsion with the lipid phosphate group that would displace PL binding, where mutation of Arg78 to alanine (R78A)

may have a more subtle effect by neutralizing the charge rather than introducing charge repulsion. We also mutated Y114 to an arginine to disfavor PC binding through charge repulsion. In addition to the structure guided mutations, we also tested a mutation previously shown to reduce PC transfer (R120H)[33,34].

Protein complementation assays in Huh7 cells, Y114R, R78E, and R120H all decrease the interaction between PC-TP and PPARδ by ~45%

**Fig. 3 | Decreased hepatic lipotoxicity in L-*Pctp*$^{-/-}$ mice on HFD. A** Decreased liver weight in KO mice fed HFD compared to WT control ($n = 4$, two-tailed $t$-test, SEM). **B** Quantification of lipid droplet area for histology ($n = 6$, two-tailed $t$-test, SEM). **C** Representative histological sections of HFD-fed KO and WT mice with H&E staining ($n = 6$). **D** Quantification of liver composition of triglycerides (TG) and free cholesterol for KO and WT mice ($n = 4$, two-tailed $t$-test, SEM). **E** Quantification of liver composition of free cholesterol for KO and WT mice ($n = 6$, two-tailed $t$-test, SEM). **F** Quantification of plasma of free cholesterol ($n = 5$, two-tailed $t$-test, SEM).

**G** Quantification of plasma total cholesterol ($n_{wt} = 6$ $n_{ko} = 5$, two-tailed $t$-test, SEM). **H** Quantification of plasma phospholipid (PL) ($n_{wt} = 5$ $n_{ko} = 4$, two-tailed $t$-test, SEM). **I** Quantification of plasma non-esterified fatty acid (NEFA) ($n = 5$, two-tailed $t$-test, SEM). **J** Quantification of plasma composition of triglyceride (TG) ($n_{wt} = 5$ $n_{ko} = 6$, two-tailed $t$-test, SEM). **K**, **L** Amount of cholesterol and TG packaged into lipoprotein particles as determined by size exclusion chromatography ($n = 3$, two-tailed $t$-test, SEM). Source data are provided as a Source Data file.

relative to wild-type PC-TP. The R78A mutation has a lesser effect, maintaining ~85% interaction compared to WT PC-TP (Fig. 7F). Using in-cell luciferase reporter assays, we assessed the ability of these mutants to suppress PPARδ-driven transactivation. In agreement with our nano-PCA data, mutants that retained PC binding demonstrate similar levels of repression to that of WT PC-TP, while mutations that attenuate the PC binding have an attenuated affect (Fig. 7G).

## Discussion

Our transcriptomic analysis of livers isolated from the previously established *Pctp*$^{-/-}$ mouse fed either normal diet (ND) or methionine and choline depleted (MCD) diet demonstrates a clear effect of both the knockout and MCD diet on PPAR activity. This analysis also suggests that PC-TP regulation of PPARδ is altered depending on diet, as MCD led to a more dramatic change in the transcriptome compared to ND fed mice. Pathway analysis comparing the effect of diet within each genotype showed changes in PPARδ controlled processes for both CHEA and KEGG analysis that was not present in *Pctp*$^{-/-}$ mice, suggesting that differential PPARδ regulation in response to dietary changes requires the presence of PC-TP. Given that PPARα and PPARδ share a highly conserved DNA-binding element and have differential occupancy at overlapping gene targets depending on nutrient status, ontology analysis may be skewed to show PPARα as a result of being the more thoroughly studied member of the PPAR family[26].

Hepatic *Pctp* deletion resulted in increased insulin and glucose sensitivity with decreases in weight gain, and lipid accumulation in the liver and skeletal muscle under high-fat diet feeding. This change occurred in absence of any significant alteration in food or water intake. Our characterization of the L-*Pctp*$^{-/-}$ showed that PC-TP deletion not only regulated liver health but also had an effect on the muscle lipid content in line with previous reports of a hepatic PPARδ-PC-muscle axis. In the liver, PPARδ controls autophagy and metabolism in the fed state and inflammatory response[35,36]. However, many of the beneficial metabolic effects associated with activation of hepatic PPARδ has been ascribed to its role in modulating liver-skeletal muscle communication. This phenomenon is regulated by a specific PC (PC (18:0/18:1)) known to be generated in the liver by processes transcriptionally controlled by PPARδ[25]. In the muscle, PC 18:0/18:1 activates PPARα, culminating in increased fatty acid uptake and utilization. The consequences of this liver-skeletal muscle axis controlled by PPARδ is increased endurance during exercise, sensitization to glucose and insulin, and a decrease in lipid content in both the liver and the muscle[25,35].

While much of our mouse model phenocopied aspects of hepatic PPARδ overexpression, we did not observe a characteristic increase in fatty acid oxidation as indicated by the respiratory ratio, or energy expenditure. However, we observed a decrease in lipid accumulation in all tissues as determined via EchoMRI and lipid profiling of liver, muscle and serum. This global decrease in lipid accumulation indicates that fatty acid homeostasis has been altered in ways that may mitigate hepatic lipotoxicity.

We show using both nano-PCA and BRET, that PC-TP and FABP5 interact directly with PPARδ in cells and that the PPARδ LBD is sufficient for this interaction in vitro as measured by TRIC. In cell, both PC-TP and FABP5 interact with intact PPARδ greater than the isolated LBD; however, this could be partly explained by differences in PPARδ LBD

localization since it lacks a nuclear localization sequence. In cell, PPARδ-LTP interaction may also be stabilized by PTMs, endogenous ligands or other interaction partners that are not recapitulated with the bacterially expressed and purified proteins. Combining these observations with luciferase reporter and qPCR assays, we show PC-TP negatively regulates PPARδ mediated transactivation. We hypothesized that ligand binding and possibly transfer are required for productive interaction between PC-TP and PPARδ. This hypothesis is supported by our mutants aimed at altering PC binding (R78A, R78E, Y114R) and transfer (R120H), as these mutants attenuated the interaction and the complimentary repressive phenotype. Serum starvation increased formation of the PC-TP–PPARδ complex and enhanced PPARδ repression suggesting that this interaction is sensitive to nutrient availability. Serum starvation had the opposite effect on the FABP5–PPARδ interaction. This result suggests that these two LTPs play opposing roles in modulating PPAR function. Taken together our data points to a role of PC-TP in regulating liver health via modulating PPARδ in a ligand-dependent manner. Further studies investigating which ligand modulates this interaction as well as the effect of these ligands on PC-TP/ PPARδ biology could shed light on the role of this complex in the pathogenesis of obesity.

The STARd family of proteins are characterized by their unique fold and ability to bind and transfer hydrophobic ligands. Members of this family range in complexity from the minimal start domain (e.g., PC-TP) to multidomain proteins, such as STARD14 (THEM1). In these complex multidomain proteins, the start domain acts as a sensor that is capable of relaying ligand-binding information to other domains to tune activity[37]. PC-TP is capable of relaying ligand-binding information to control the activity of interacting proteins such as THEM2 or PAX3[28]. PC-TP increases the thioesterase activity of THEM2, which antagonizes fatty acid shuttling into the mitochondria by cleaving activated fatty-acyl-CoA species into NEFA[2,6]. Interestingly, knockout of PC-TP, but not THEM2, increases the expression of PPAR target genes, suggesting repression of PPAR activity is PC-TP specific[7].

Our data suggests that PC-TP interacts with PPARδ and regulates transcriptional activation in response to alterations in diet (Fig. 8). Further work is required to characterize the consequence of the PC-TP–PPARδ signaling in other tissues as well as the role of other START domain-containing proteins ability to modulate PPARs, and other lipid-binding nuclear receptors.

## Methods

### Reagents

Buffers and reagents for crystallography were purchased from Sigma, Fisher, Polysciences, or Cayman, Inc. The pMCSG7-His plasmid was a gift from John Sondek (University of North Carolina, Chapel Hill), whereas Nano-PCA and BRET vectors were graciously given by Dr. Haian Fu (Emory University, Atlanta, GA).

### Expression of PC-TP, FABP5 and PPARδ LBD, and FL-PPARδ

Following sequencing, vectors containing coding regions of either the PPARδ ligand-binding domain (LBD) in pRSET vector, Full-length PPARδ in PSMT3 vector or FABP5 /PC-TP in pMCSG7 were transformed into BL21 DE3 *E. coli*. Cultures were grown in terrific broth (TB; 6 × 1.3 L) at 37 °C with gentle shaking to an OD$_{600}$ of 0.6, cooled to 18 °C, induced with 0.5 mM isopropyl-1-thio-D-galactopyranoside (IPTG), and

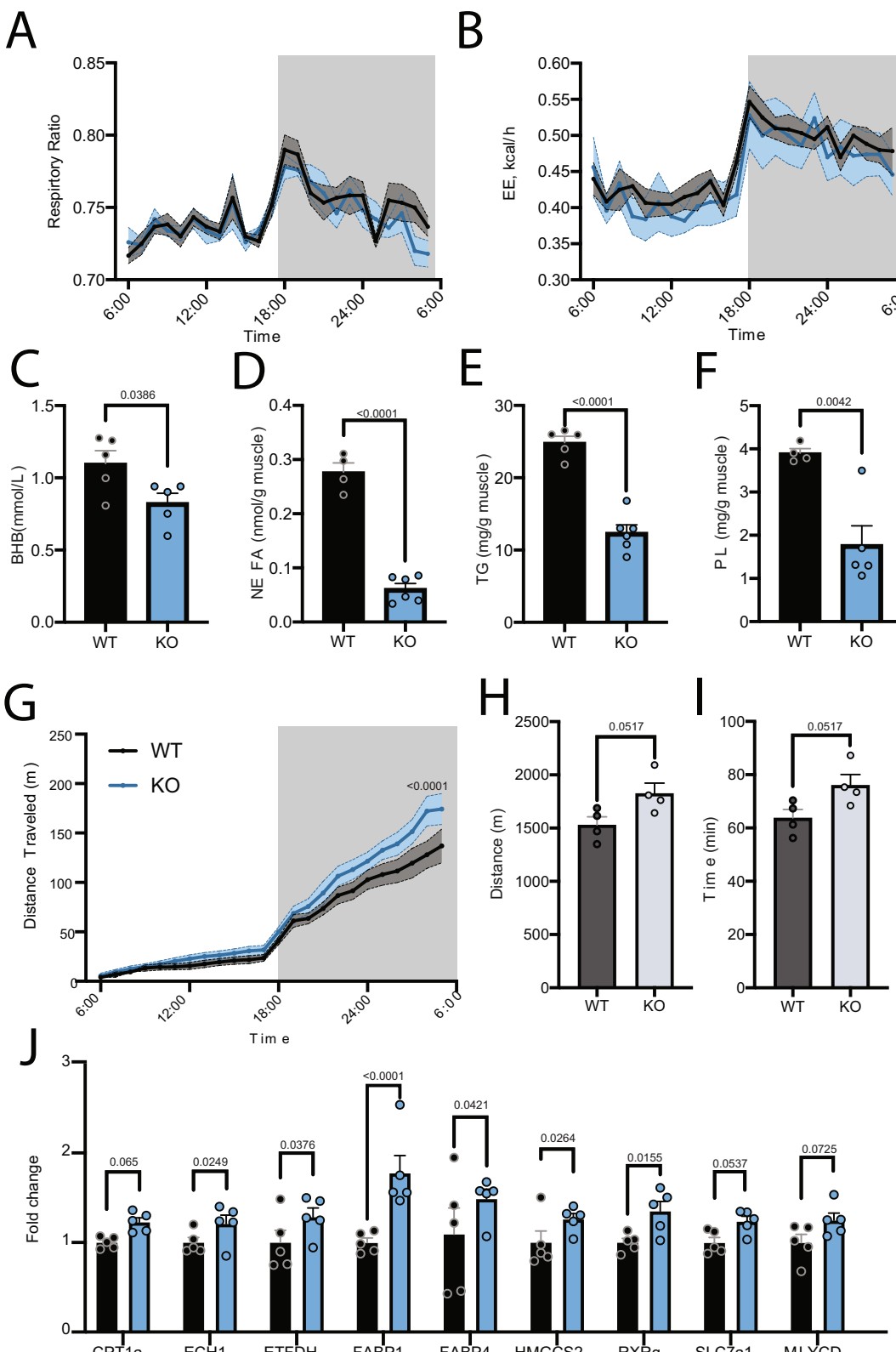

**Fig. 4 | L-*Pctp*⁻/⁻ drives beneficial metabolic alterations through muscle-liver axis. A**, **B** Promethion cage tracking of the respiration ratio and energy expenditure (*n* = 6, two-tailed *t*-test, SEM). **C** Quantification of plasma beta-hydroxybutyrate (BHB) levels using ELISA (*n* = 5, two-tailed *t*-test, SEM). **D** Quantification of muscle non-esterified fatty-acid (NEFA) content ($n_{wt}$ = 4 $n_{ko}$ = 6, two-tailed *t*-test, SEM). **E** Quantification of muscle triglyceride (TG) content (*n* = 5, two-tailed *t*-test, SEM). **F** Quantification of muscle phospholipid (PL) content ($n_{wt}$ = 4 $n_{ko}$ = 5, two-tailed *t*-test, SEM). **G** Promethion cage of free running for HFD L-*Pctp*⁻/⁻ mice show a trend toward increased movement at later time points (*n* = 6, two-tailed *t*-test, SEM). **H**, **I** Quantification of endurance stress test of WT PC-TP KO mice (*n* = 6, two-tailed *t*-test, SEM). **J** Relative gene expression in the livers of WT and KD mice fed HFD shows significant alteration in several genes known to be regulated by PPARs. # denotes *p*-value < 0.07 but >0.05 (*n* = 5, two-way ANOVA, SEM). Source data are provided as a Source Data file.

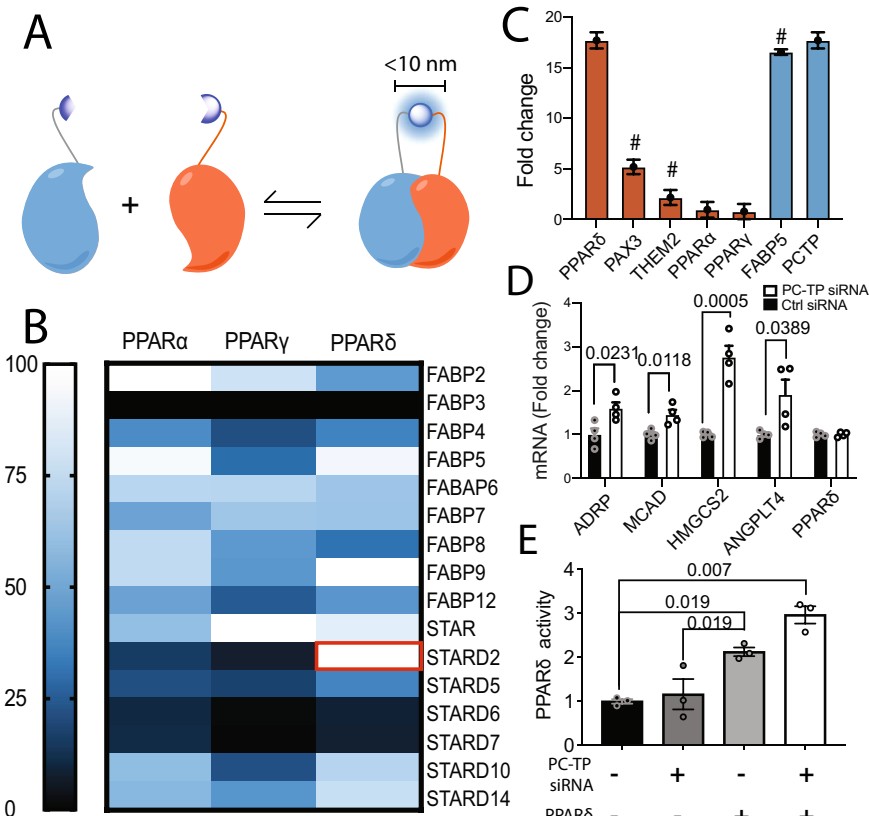

**Fig. 5 | Discovery of a repressive PC-TP–PPARδ complex. A** Depiction of the nano-PCA assay. **B** Heat map of HEK293T cells were transfected with portions of luciferase fused to LTPs or PPARs, interactions for each PPAR isoform were normalized to the highest interaction ($n = 1$). Calculated $z'$ and $S/B$ noise suggest that this assay is robust using Them1-Them1, a known trimer, as a positive control and beta-catenin with FABP3 as a negative control ($z' = 0.795$ $S/B = 7.03$). **C** Bar graph showing the quantification from the nano-PCA screen. PPARδ (blue) interacts with PC-TP (orange) stronger than known partners (#) of either protein ($n = 1$ biological replicate, 4 technical replicates, SEM). **D** To evaluate the effect of PC-TP on the activity of PPARδ, siRNA knockdown of PC-TP was performed in Huh7 cells using targeted and scrambled siRNA. Knockdown (KD) efficiency was confirmed by western blot. Target mRNA expression levels at 48H post KD was quantified by qRT-PCR and showed increased PPARδ transactivation when PC-TP is transiently KD ($n = 4$, one-way ANOVA, SEM). **E** Luciferase reporter assay in conjunction with siRNA of PC-TP's confirms PC-TP negatively regulates PPARδ's transactivation ($n = 3$, one-ANOVA, SEM). Source data are provided as a Source Data file.

allowed to grow overnight. 0.01% soy lecithin was added to PC-TP cultures at induction to help increase protein yield. Similarly, 100 mM $ZnSO_4$ was added to growths of full-length PPARδ to aid in correct folding of zinc finger domain. The next morning cells were spun down at $3500 \times g$ for 20 min. Pellets were frozen at −80 C until they were purified.

### Purification of PC-TP and FABP5

Pellets were thawed, homogenized, and lysed in 150 mM NaCl, 20 mM Tris–HCl pH 7.4 supplemented with 5% glycerol, 25 mM imidazole, 5 mM β-mercaptoethanol, 100 μM PMSF, DNAse A, and lysozyme. The lysate was then spun down at $18,000 \times g$ for 60 min the supernatant was isolated and loaded onto a immobilized metal affinity chromatography (IMAC) column with the following buffers: buffer A (150 mM NaCl, 20 mM Tris, 20 mM imidazole, 5% glycerol, pH 7.5) and buffer B (150 mM NaCl, 20 mM TRIS 250 mM imidazole, 5 % glycerol, pH 7.5). Fractions containing PC-TP or FABP5 were then pooled, run on a superdex 75 pg 16/60 size exclusion column (20 mM HEPES, pH 8.2, 200 mM NaCl, 0.05% tween-20 and 0.5 mM TCEP), and frozen at −80 °C.

### Purification of PPARδ

PPARδ was expressed in *E. coli.* and purified as previously described[38]. Briefly, growth conditions and pellet processing were identical to PC-TP, except PPARδ pellets were lysed with a buffer containing 50 mM HEPES, pH 8.0, 500 mM ammonium acetate, 20 mM imidazole, 1%

triton x100, 10% [v/v] glycerol and 10 mM β-mercaptoethanol. IMAC purification was performed using buffer A (50 mM Tris–HCl, pH 8.0, 500 mM ammonium acetate, 20 mM imidazole, 10% [v/v] glycerol and 10 mM β-mercaptoethanol) and buffer B (50 mM Tris–HCl, pH 8.0, 500 mM ammonium acetate, 250 mM imidazole, 10% [v/v] glycerol and 10 mM β-mercaptoethanol). The Full-length PPARδ was further purified using ion affinity chromatography. Briefly, following IMAC column fractions containing FL-PPARδ were diluted to a salt concentration of -150 mM and ran against tandem Capto S and Capto Q columns. Both full-length PPARδ and the LBD were then further purified using exclusion chromatography via Superdex 75 pg 16/60 with a running buffer containing 20 mM HEPES, pH 8.3, 200 mM NaCl, 0.05% tween-20 and 0.5 mM TCEP.

### Protein crystallization, data collection, and structure determination

PC-TP was concentrated to 15 mg/mL. Crystals were grown by hanging drop vapor diffusion at 18 °C in drops containing 1 μL of PC-TP and 1–2 μL of well buffer containing 3.4–3.8 M sodium formate and 0.1 mM sodium acetate pH 5.7. Crystals grew rapidly, often with significant growth overnight. Crystals were cryoprotected in well solution containing 15% glycerol and flash frozen in liquid nitrogen. Data were remotely collected from the Southeast Regional Collaborative Access Team (SER-CAT) at the Advanced Photon Source (APS), 22ID beamline (Argonne National Laboratories, Chicago, IL). Data were processed and scaled using HKL-2000[39] and phased by molecular replacement using

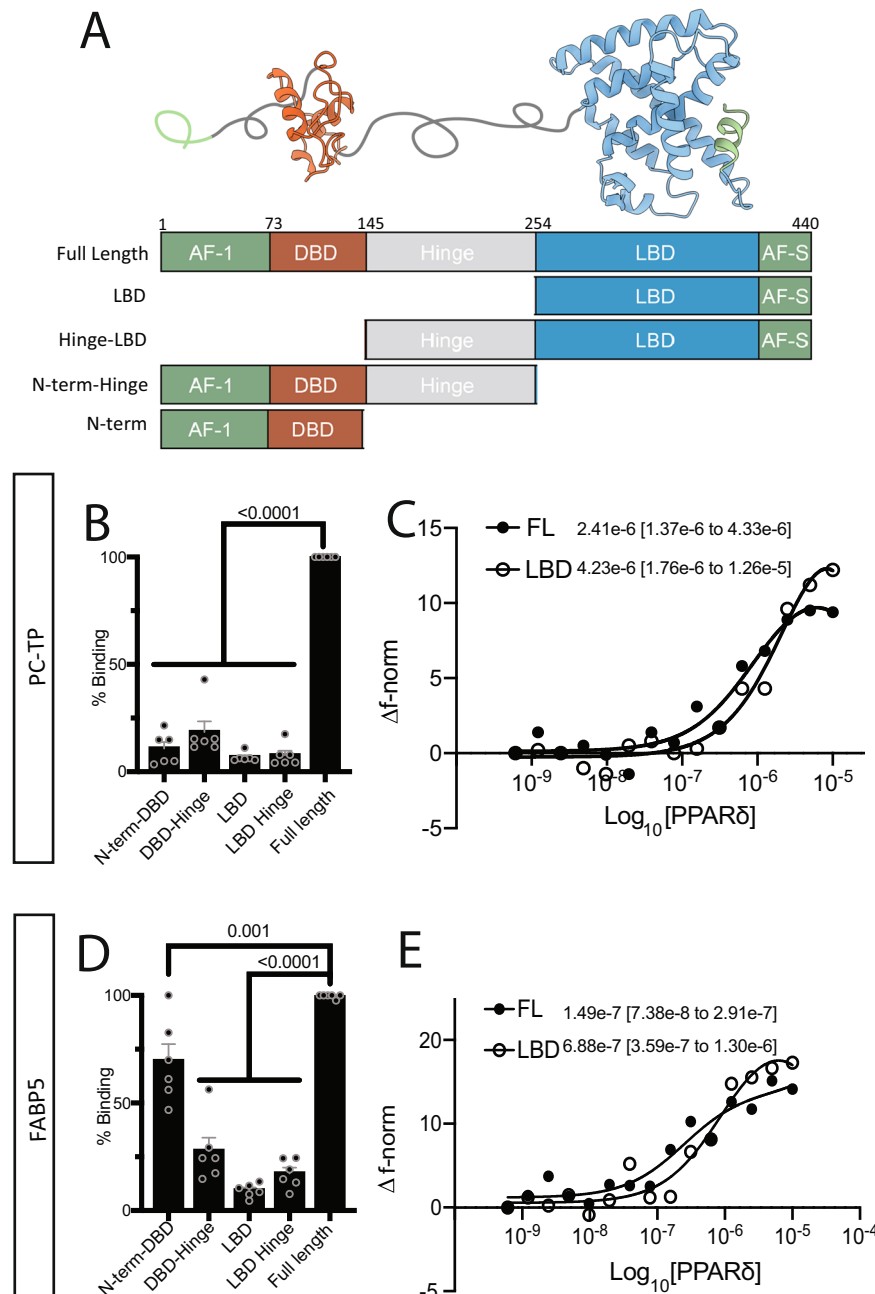

**Fig. 6 | Mapping the PC-TP/FABP5–PPARδ interacting domains. A** Homology model of full-length PPARδ generated using the structure of full-length PPARγ. Schema of the modular structure of PPARδ with representative domain truncations used in subsequent PCA experiments **B**, **D** To determine the locus of PPARδ interaction with PC-TP or FABP5, respectively, we utilized our nano-PCA in Huh7 cells ($n = 6$, one-way ANOVA, SEM). **C**, **E** TRIC experiments on 50 nm Cy5 NHS labeled PC-TP or FABP5 titrating in PPARδ Full-length (FL) or ligand-binding domain (LBD) ($n = 3$). Source data are provided as a Source Data file.

Phaser-MR[40] using a previously published PC-TP structure as a reference model (PDB: 1LN1)[31]. Models were built using COOT[40,41] and refined using PHENIX and PDB_REDO[42]. Structures were visualized using PyMOL (Schrödinger, LLC).

**Cloning and mutagenesis**

Full-length, wild-type human PC-TP (residues 1–220) was subcloned into pDONR201. Mutants R78E, R78A, Y114R, and R120H mutants were established in pDONR201 prior to subcloning into pDEST26 for luciferase reporter or nano-PCA, respectively. Similarly, wild-type human PPARδ (residues 1-470) was subcloned into pDONR223. PPARδ truncations were first cloned into pDONR201 using BP clonase and subsequently cloned into nano-PCA vector via LR clonase. All

mutagenesis was performed using NEB Q5 site directed mutagenesis kit (New England biosciences).

**In vitro-binding assays**

PPARδ LBD or FL-PPARδ was purified in assay buffer (20 mM Hepes pH 8.3, 200 mM NaCl, 5% glycerol, 0.05% tween-20, 0.5 mM TCEP) and labeled using the NHS-CY5 dye as per companies instructions[29]. Proteins were labeled with a Cy5 fluorophore through covalent linkage to lysine amines using the Monolith NT.115 Protein Labeling Kit RED-NHS (NanoTemper Technologies, München, Germany). LTPs (10 μM–610 pM) were then incubated with 50 nM Cy5-labeled FL-PPARδ or PPARδ LBD overnight at 4 °C before assessing binding. TRIC measurements were taken using a Dianthus NT.23 Pico (NanoTemper

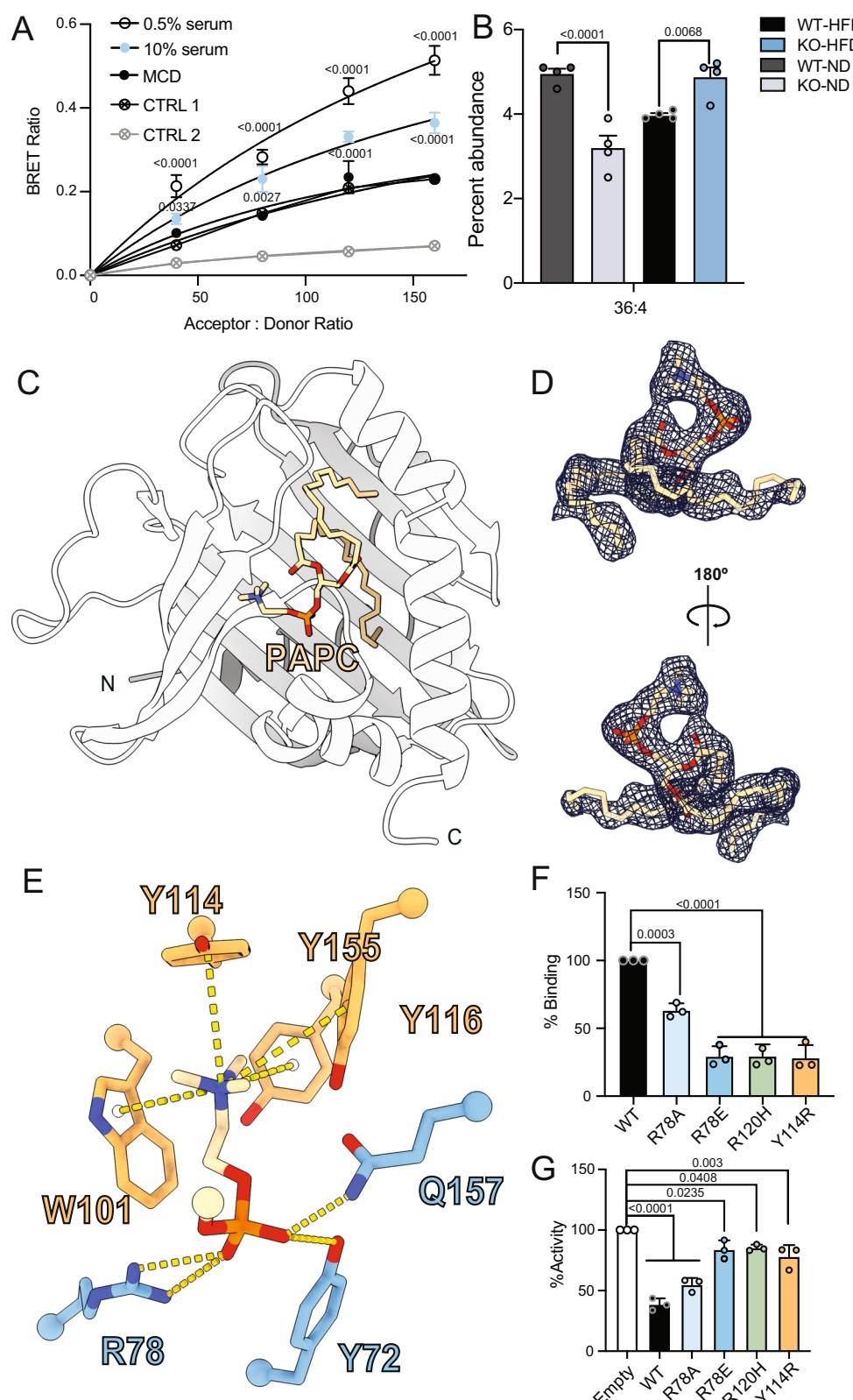

Technologies) instrument. Data for three independent measurements were fitted with a non-linear regression model in GraphPad Prism 8.0.

## PC-TP knockdown

SMARTpool siRNA against human PC-TP (catalog ID: L-013786-02-0050) and scrambled control siRNA were purchased from Dharmacon. Huh7 cells were seeded at 35,000 cells per well in 24-well plates and

allowed to adhere overnight. Cells were transfected with siRNA using Dharmafect transfection reagent 4, at a final concentration of 25 nM siRNA and 0.6 µL transfection reagent per well. Cells were harvested 24 h after transfection in Trizol reagent (Invitrogen), and RNA was purified using the RNAeasy kit (Qiagen). RNA was reversed transcribed into cDNA using the High-Capacity RT kit (Applied Biosystems) prior to analysis by qPCR as described below.

**Fig. 7 | Lipid dependence of PC-TP–PPARδ interaction. A** BRET data performed in Huh7 cells confirms the PC-TP- PPARδ complex formation and a slight increase in serum starved Huh7 cells ($n = 3$, two-way ANOA, SEM). CTRL 1 corresponds to nano-Luciferase (nLuc)-PC-TP with empty Venus, CTRL 2 corresponds to Venus-PPARδ with empty nLuc. **B** Lipidomic profiling of the PC species isolated from ERs of livers from $Pctp^{-/-}$ on normal diet (ND) or high-fat diet (HFD) reveals changes in PC 36:4 (PAPC) ($n = 4$, two-way ANOVA, SEM). **C** 2.18 Å structure of PC-TP in complex with PC 36:4 (PAPC) used to design proposed mutations to alter PC-TP PC binding. **D** Polder map (sigma = 3) showing ligand density associated with PC 36:4 (PAPC) bound to PC-TP. **E** This structure suggests PC-TP selectivity for PCs is driven via aromatic residues that form a cage surrounding the choline head group (shown in melon) and residues that coordinate the phosphate backbone (shown in baby blue) (cutoff of 3.5 Å). **F** To determine the requirement of PC-TP PC binding on the interaction with PPARδ, nano-PCA was performed in Huh7 cells cultured in 0.5% serum ($n = 3$, one-way ANOVA, SEM). **G** To determine if mutants retain the ability to inhibit PPARδ, we performed luciferase reporter assays in Huh7 cells cultured in serum starvation conditions suggesting that reducing PC-TP PC binding or transfer ablates the ability to suppress the activity of PPARδ ($n = 3$, one-way ANOVA, SEM). Source data are provided as a Source Data file.

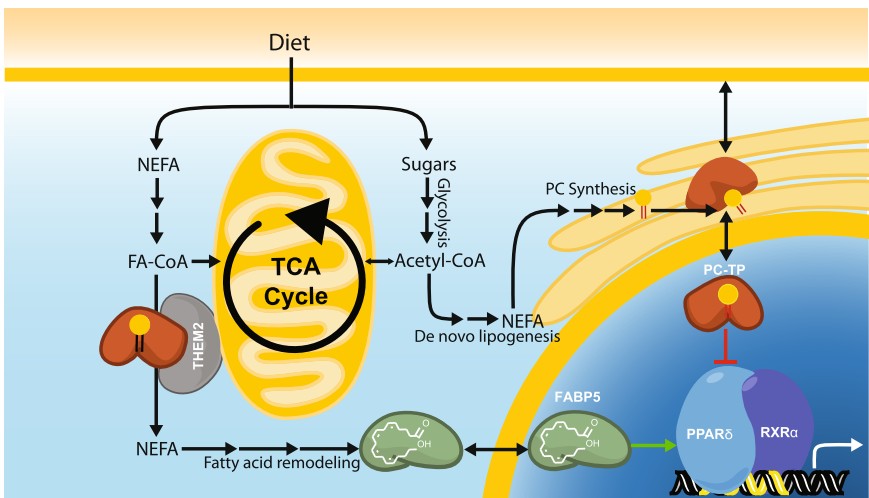

**Fig. 8 | Cellular context for a LTP-PPARδ axis.** Here, we outline the mechanism by which PC-TP may regulate PPARδ activity. Diet supplies a myriad of nutrients from carbohydrates to lipids. Fats from diet are transported throughout the body in lipoprotein particles. Once liberated by various lipases these non-esterified fatty acids (NEFA) are then absorbed by cells where they are either used as an energy source, incorporated into other fatty acid species, or remodeled into signaling moieties such as eicosanoids, leukotriene, and thromboxane. Sugars are broken down in the cytosol via glycolysis, the products of which can be utilized as building blocks for a number of macromolecules or further metabolized in the mitochondria to produce more energy. FABP5 (Shown in green) binds polyunsaturated fatty acids, derived from diet, which can lead to nuclear localization where it increases PPARδ transactivation of genes. PC-TP (shown in orange) regulates membrane fluidity via the transfer of PCs synthesized in the ER, metabolism through increasing the thioesterase activity of THEM2(shown in gray), and transcription through its interaction with PPARδ, characterized here. Under high-fat, high carbohydrate diets, excess nutrients are stored in glycogen and lipids including PCs. PCs generated via de novo lipogenesis contain medium chain saturated fatty-acyl-chains. Whereas essential fatty acid obtained from diet contain unique desaturation states, capable of altering FABP5 localization. We hypothesize that these uniquely generated PCs species can perturb the homeostatic regulation of PPARδ culminating aspects of metabolic syndrome.

For siRNA experiments with luciferase reporter assays, Huh7 cells were seeded at 7000 cells/ well in clear-bottomed, white-welled 96-well culture plates in DMEM F12 medium supplemented with 10% charcoal-stripped FBS and allowed to adhere overnight. Cells were transfected with the siRNA described above (25 nM) together with a luciferase reporter expressing firefly luciferase under the control of a PPAR-response element (100 ng) and a constitutive luciferase reporter expressing Renilla luciferase under the control of the CMV promoter (20 ng). Some cells were also transfected with full-length human PPARδ in a pSG5 vector (100 ng). Transfections used Dharmafect Duo transfection reagent (0.2 μL per well). Luciferase signal was measured 24 h after transfection using the Dual-Glo kit as described below.

**In-cell activation assays**
Huh7 cells were grown and maintained in DMEM F-12 containing L-glutamine, sodium pyruvate, and phenol red (Invitrogen) supplemented with 10% FBS (Invitrogen). Cells were transferred into a 96-well plate at a density of 7000 cells/well and allowed to grow for one day prior to transfection. One hundred ng/well pSG5 vector harboring full-length human PPARδ receptor, 100 ng/well PPAR- response element-driven firefly luciferase reporter (PPAR-response element X3-TK-luc), and 20 ng/well constitutive Renilla luciferase reporter (phRLtk) in the presence or absence of 25 ng/well wild-type or mutant variant human PC-TP cloned into the pDEST26 vector was added to FuGENE HD in

Opti-MEM (Invitrogen). Four microliters of this solution was used to transfect 60−70% confluent cells overnight. Twenty-four hours post transfection, media was changed to fresh culture media, DMEM F-12 containing either 10% serum or 0.5% serum and assayed with Dual-Glo luciferase substrate (Promega). Cells that were starved of serum were grown in culture media containing 0.5% FBS for 24 h before reading. Firefly activity was divided by Renilla activity to account for cell number, viability, and transfection efficiency.

**RNAseq analysis and quantitative PCR**
Total RNA was isolated from liver (10 μg) or cells using miRNeasy (QIAGEN, Hilden, Germany) following manufacturer's instructions. cDNA was synthesized with a High-Capacity cDNA Reverse transcription Kit (Applied Biosystems, Foster City, CA, USA). RNA quality was assessed via agarose gel and nanodrop prior to library preparation and transcriptome sequencing was conducted by Novogene Co., LTD (Beijing, China). Genes with adjusted $p$-value < 0.05 and |log2(Fold-Change)| > 0 were considered as differentially expressed. Relative mRNA expression was also determined by quantitative PCR using SYBR Green Real-Time PCR Master Mix (QIAGEN, Hilden, Germany). Equal amounts of cDNA samples were subjected to qPCR using the StepOnePlus (Applied Biosystems, Foster City, CA, USA) in a 96-well plate. mRNA isolated from cells were analyzed relative to Thyroxine-binding globulin (TBG), whereas, arrays were analyzed per manufactures

instruction[43]. Qiagen RT2 custom arrays (CLAM28936C) layout and information is presented in Supplemental Table 2. The nucleotide sequences of oligonucleotides used for qPCR are presented in Supplemental Table 3.

## In-cell protein complementation assays

Huh7 cells were maintained in DMEM F-12 containing 10% FBS. Cells were then transferred into a 96-well plate at a density of 7000 cells/well and grown for 24 h. For nano-PCA assays, cells were then transfected with one hundred ng/well of WT or mutant human PC-TP or FABP5 in the split nluc-PCA vector, one hundred ng/well of WT or mutant PPARδ were added to FUGENE HD in Opti-MEM. Twenty-four hours post transfection, media was changed to either fresh culture media, DMEM F-12 containing 0.5% serum and assayed using nano-glo luciferase substrate (Promega). Controls include empty nluc-PCA vector mixed with either LTP or PPARδ to account for nonspecific interaction. Signal from the interaction was normalized to the highest signal from a negative control. For BRET assays, cells were transfected with the designated ratio of donor: acceptor, using a 5 ng of nluc fusion protein. These mixtures were then added to FUGENE HD in Opti-MEM. Twenty-four hours post transfection, media was changed to either fresh culture media, DMEM F-12 depleted of methionine and choline, or DMEM F-12 containing 0.5% serum.

## Animals and diets

We have ensured that all animal research present within the manuscript comply with the Institutional Animal Care and Use Committee of Weill Cornell Medical College. Mice were monitored by daily health status observation by technicians supported by veterinary care. Housing and husbandry were conducted in facilities with a sentinel colony health monitoring program and strict biosecurity measures to prevent, detect, and eradicate adventitious infections. Tissue-specific knockdown mice were created by a LoxP/Cre system C56BL6 background mice with *Pctp* flanked by two LoxP sites (*Pctp*flox/flox) using a Crispr/Cas9 set up (Supplemental Fig. 7). *Pctp*flox/flox were transduced with AAV8 harboring a vector for Cre recombinase driven by the TBG promoter to generate hepatocyte-specific deletion of *Pctp*−/− (*L-Pctp*−/−). Similarly, control mice were treated with equivalent titer of empty AAV8. These mice were viable and displayed no apparent developmental abnormalities. Knockdown efficiency was assayed via western blot, comparing liver and kidney isolated from KO and control mice. Mice were housed in a barrier facility on a 12 h light/dark cycle with free access to water and diet. Male mice were weaned at 4 weeks of age and fed chow (PicoLab Rodent Diet 20; LabDiet, St. Louis, MO, USA). Alternatively, 5 weeks old male mice were fed a HFD (D12492: protein 20% kcal, fat 60% kcal, carbohydrate 20% kcal, energy density 5.21 kcal/g; Research Diets Inc., New Brunswick, NJ, USA) for 12 w. Following 6 h fasting (9:00 AM to 3:00 PM) or 18 h fasting (6:00 PM to 12:00 PM), 17-week old mice were euthanized and plasma was collected by cardiac puncture. Tissues were harvested for immediate use or snap frozen in liquid nitrogen and stored at −80 °C.

## Endurance stress test

Mice were acclimated to treadmill running tracks incrementally for 5 days starting from 10 min at 0.3 km/h to 60 min at 1.2 km/h. To determine maximal exercise capacity, treadmill speed was started at 0.3 km/h and increased 0.3 km/h every 3 min until the speed reached 1.2 km/h. The speed was kept constant (1.2 km/h) until mice reached exhaustion.

## Metabolic monitoring

Mice were housed in individual cages for 1 week for acclimation prior to metabolic monitoring[44]. Mice were then housed in temperature-controlled cabinets with a 12 h light/dark cycle and monitored using the Promethion Metabolic Screening System (Sable Systems International, North Las Vegas, NV). Rates of $O_2$ consumption ($VO_2$) and $CO_2$ production ($VCO_2$) were determined at 5 min intervals. Values of respiratory exchange ratio (RER) were calculated as $VCO_2/VO_2$. After 2 days acclimation, metabolic parameters were recorded over 24 h with or without a running wheel inside the cage. Physical activities and voluntary running on wheel were determined according to beam breaks within a grid of photosensors built outside the cages. Energy expenditure was calculated by indirect calorimetry and adjusted by ANCOVA using VassarStats to adjust for differences in lean body mass using lean body composition determined by magnetic resonance imaging EchoMRI (EchoMRI, Houston, TX).

## Tolerance tests

Glucose tolerance tests (GTT), insulin tolerance tests (ITT), and pyruvate tolerance tests (PTT) were performed with minor modifications. In brief, mice were fasted 6 h for GTT, 4 h for ITT, and overnight (16 h) for PTT. A drop of blood from the tail tip was subjected to glucose measurement at baseline and at regular intervals using GE 100 Blood Glucose Monitor (General Electric, Ontario, CA). Glucose solution was administered by oral gavage with 1.5 g/kg for *L-Pctp*−/− mice and their littermate controls. Insulin solution was administered by intraperitoneal injection with 0.75 U/kg for *L-Pctp*−/− mice, and littermate controls. Pyruvate solution was administered by gavage with 2 g/kg for *L-Pctp*−/− mice and littermate controls.

## Immunoblot analysis

Immunoblot analysis were performed by standard techniques. Briefly, tissues or cells were homogenized in a RIPA buffer containing cOmplete Protease Inhibitor Cocktail and PhosphoSTOP Phosphate Inhibitor Cocktail Tablets (Roche, Indianapolis, IN, USA) using a Bead Ruptor 24 Elite bead mill homogenizer (Omni International, Kennesaw, GA, USA). Protein concentrations were determined by using a BCA reagent (Thermo Fisher Scientific, Springfield Township, NJ). Equal amounts of protein samples were separated by SDS-PAGE and transferred to nitrocellulose membranes for immunoblot analysis. Membranes were incubated with primary antibodies overnight and probed with respective secondary antibodies (Dako-Agilent, Santa Clara, CA, USA) for 1 h. Polyclonal antibodies to PC-TP and Them2 were prepared as previously described[28,45] (dilution, 1:1000). GAPDH antibody was from Novus Biologicals, Inc. (Catalog # NB100-56875) (dilution, 1:1000). Polyclonal goat anti-rabbit immunoglobulins from Agilent Dako (Catalog # P0448) (dilution, 1:5000) was applied as secondary antibody. Bands were then developed using enhanced chemiluminescence (SuperSignal West DURA, Thermo Fisher Scientific, Springfield Township, NJ, USA) and imaged by ChemiDoc XRS + (BioRad, Hercules, CA, USA).

## Histopathology

Freshly harvested tissues from 17-week old mice were immersed in 10% neutralized formaldehyde for 2 days. Following paraffin embedding, sectioned tissues were stained with Hematoxylin and Eosin by the Laboratory of Comparative Pathology, Memorial Sloan Kettering Cancer Center, NY, USA. Slides were visualized using an Eclipse Ti microscope (Nikon, Tokyo, Japan).

## Tissue triglyceride concentrations

Lipids were extracted from frozen specimens with a mixture of chloroform/methanol (2:1) using Folch's method as previously described[46]. Concentrations of triglycerides were assayed enzymatically (FUJIFILM Wako Diagnostics, Mountain View, CA).

## Plasma assays and fast protein liquid chromatography

Enzymatic assay kits were used to measure plasma concentrations of triglycerides, free fatty acids, total cholesterol, free cholesterol, phospholipid (FUJIFILM Wako Diagnostics, Mountain View, CA, USA),

ß-hydroxybutyrate (Stanbio Laboratories, Boerne, TX). Plasma concentrations of insulin were measured by using an ELISA kit (Crystal Chem, Downers Grove, IL), according to the manufacturer's protocol. Equal volumes of plasma were pooled from five mice and lipoproteins were fractionated by fast protein liquid chromatography (ÄKTA pure FPLC system, GE Healthcare, Pittsburgh, PA, USA) and triglyceride and cholesterol in fractions quantified with reagent kits (Wako Diagnostics, Mountain View, CA).

### Lipidomic analysis on isolated endoplasmic reticulum

To characterize phospholipids in mouse liver endoplasmic reticulum (ER), ER was first isolated by density centrifugation according to a Nature Protocols method[47]. Briefly, 500 mg mouse liver was homogenized in 1 mL ice cold PBS, pH 7.4 using a glass, mortar and pestle style homogenizer. The homogenate was then centrifuged for 5 min at $740 \times g$ and the pellet was discarded. The resulting supernatant was collected and further centrifuged for 10 min at $9000 \times g$. The pellet, containing crude mitochondria was discarded, and the supernatant, which contains the ER was centrifuged for 30 min at $20,000 \times g$. In the final step, the resulting supernatant was centrifuged for 1 h at $100,000 \times g$ to pellet purified ER. All centrifugation steps were performed at 4 C. Lipids were then extracted from the ER fraction. For this, the pellet was resuspended in 1 mL 2:1 methanol:chloroform (v/v). The sample was then mixed in a stand vortexer for 15 min. To separate phases and aid in the partition of zwitterionic lipids to the organic phase, 0.5 mL 0.1 mM NaCl was added. The aqueous phase was removed and the organic phase was retained then subsequently dried under gentle nitrogen stream. Extracted lipids were reconstituted in 500 μL 1:1 methanol:chloroform.

Targeted lipidomics was performed using a Sciex QTrap 3000 (Framingham, MA, USA), whereby lipids were directly infused into the mass spectrometer using a syringe pump at a flow rate of 5 μL/min. Instrumental parameters; electrospray voltage (−3500 kV), collision energy (−30 eV), and declustering potential (−70 arb units) optimized using analytical grade standards, di17:0 phosphatidylcholine (PC) and phosphatidylethanolamine (PE), both purchased from Avanti Polar Lipids. The distribution of lipids in ER was determined by shotgun lipidomics. For this, phosphatidylcholine species were identified by conducting precursor ion scanning for $m/z$ 184 in the positive ion mode, which corresponds to the mass of the phosphocholine head group. Similarly, PE was detected by precursor ion scanning for $m/z$ 196 in the negative ion mode. For each precursor ion scan, resulting peaks with signal to noise ratio greater than 5 were fragmented for characterization. However, quantification of identified species was achieved using the area under the curve from precursor ion scans. A phospholipid profile was then generated and used to establish differences in distributions.

### Untargeted lipidomics

**Quality control and internal standards.** Pooled quality control samples were prepared by aliquoting 5 μL of each liver extract into a single vial. This sample was spiked with 10 μL internal standard (Splash Lipidomix, Avanti Polar, Birmingham, AL). This analytical grade standard contains odd-chain, deuterated lipids in lipid classes and ratios present in human plasma. Pooled QC samples were run after every 10 samples as well as at the beginning and end of the analytical run. Quality control samples were used to ensure the stability of the instrument during analysis. The coefficient of variation for each identified lipid within the pooled QC was then calculated using a cutoff of <40%. The internal standard was used to optimize instrumental parameters, such as electrospray voltage, collision energy, and others; these parameters were held consistent over the course of analysis. Lipids in the internal standard were also used to monitor injection consistency from sample to sample and

additionally for signal correction/batch correction. For signal correction, the analytical signal of each identified lipid was normalized by the signal of internal standard, LPC 18:1d7.

**Extraction.** Lipids were extracted from livers using a high-throughput, monophasic, methyl t-butyl ether (MtBE)-based method. Using an automated pipetting and sample preparation system (Biotage Extrahera, Uppsala, Sweden), 50 μL of liver extract loaded into preconditioned wells containing 10 μL methanol and 10 μL of internal standard, Splash Lipidomix (Avanti Polar, Birmingham, AL). To each well, 200 μL methanol containing 50 μg/mL BHT was then added, and the sample was mixed by 3 up and down passes of the automated sample handling pipette. The samples were then centrifuged at 4000 rpm for 5 min to pellet precipitated protein. The supernatant was recovered and transferred to a separate deep well 96-well plate for extraction. To extract lipids from the supernatant, 250 μL MtBE:methanol (3:1 v/v) was added to all wells and mixed with 3 up and down passes of the automated sample handling pipette. The sample plate was then centrifuged at $1000 \times g$ for 3 min and the supernatant filtered through a 0.25 mm polytetrafluoroethylene (PFTE) filter plate (Biotage, ISO-LUTE® FILTER + , Uppsala, Sweden) The recovered extract was then dried under nitrogen gas and subsequently reconstituted to 200 microliters in acetonitrile:isopropanol (1:1 $v/v$) methanol for LC/MS analysis.

**Chromatography.** Ten microliters of extracted lipids were resolved on a Vanquish UHPLC (Thermo Scientific, Waltham, MA) using a Thermo Scientific Accucore C18 (4.6 × 100 mm, 2.6 μm) column on a 30 min linear gradient, whereby Solvent A was 60:40 acetonitrile:water and Solvent B 90:10 isopropanol:acetonitrile. Both solvents in the mobile phase contained 0.1% formic acid and 10 mM ammonium formate. The column temperature was set at 50 °C and a flow rate of 0.4 mL/min was constant throughout analysis. All chromatography parameters are shown in the table below (Supplemental Table 4).

**Mass spectrometry.** Eluted lipids were analyzed by a Thermo IDX mass spectrometer operated in both the positive and negative ionization modes successively (Thermo Scientific, Waltham, MA). For all experimental samples, a high-resolution MS scan was conducted on 120,000 FWHM resolution. To support compound identification, a data-dependent acquisition method was used on pooled liver QC samples. For this, a high-resolution MS scan was conducted on each pooled sample at 120,000 FWHM resolution and ions above instrumental noise threshold were systematically fragmented for structural elucidation. All MS/MS spectra were conducted using 30,000 FWHM resolution. Instrumental parameters used during analysis were optimized using the pooled quality control sample and the analytical grade internal standard. Parameters were held constant over the course of the analysis. A table of all instrumental parameters is recorded below (Supplemental Table 5).

**Lipid identification.** Raw mass spectral data were uploaded into LipidSearch software v4.2 (Thermo, San Jose, CA) for lipid identification. Peaks were detected and quantified using the QEX product ion search parameters, where 5.0 ppm was used as both parent and product mass tolerances. For identification of species, a representative liver sample was fragmented and its peaks used for identification only. For alignment, full scan mass spec data of each sample was aligned and a pooled QC was designated as the control. The identified features were then aligned with the full scan data from each respective patient sample using a 0.1 min retention time tolerance Lipids were annotated with LipidSearch v4.2 software (Thermo). Only MS2 level confirmed lipid species grade A, B and, C were used and lipids with grade D or lower were removed. Grade "A" calls are lipids of which fatty acid chains and class were identified completely,

grade "B" calls are lipids of which class and some fatty acid chains were identified, grade "C" calls are lipids of which class or fatty acid was identified, while low confidence "D" identifications are only matched according to mass. All preliminary identifications made by the software were manually reviewed to ensure appropriate identification and quantitation.

### Rates of fatty acid oxidation

Rates of fatty acid oxidation in muscle tissues were measured by degradation of $^{14}$C-palmitate (American Radiolabeled Chemicals; St. Louis, MO, USA) into $^{14}$C acid soluble metabolites (ASM) and $^{14}$C-labeled $CO_2$[46,48]. Briefly, gastrocnemius muscle strips were collected from 6 h fasted mice. Tissues were kept on ice no longer than 30 min. Muscle tissues were minced and homogenized in a Dounce homogenizer followed by centrifugation for 10 min at $420 \times g$. Supernatants were transferred to microtubes containing 0.4 µCi $^{14}$C-palmitate/500 µM palmitate conjugated with 0.7% fatty acid-free BSA and incubated at 37 °C for 30 min. $^{14}$C-labeled $CO_2$ produced by TCA cycle was captured onto filter paper soaked with 1 M NaOH and $^{14}$C-labeled ASM were separated with 1 M perchloric acid. $^{14}$C-labeled $CO_2$ in the filter paper and $^{14}$C-labeled ASM in the supernatant were dissolved in Ecoscint H (National Diagnostics; Atlanta, GA, USA) and were counted using a liquid scintillation counter (Beckman Coulter; Danvers, MA).

### Hepatic triglyceride secretion rates

Following a 12 h fast, the lipoprotein lipase inhibitor Tyloxapol (500 mg/kg of body weight) (Sigma-Aldrich) was administered through retro-orbital injection[46]. Tail tip blood samples (25 µL) were collected into microtubes before Tyloxapol injection and at regular intervals for up to 2 h. Serum triglyceride concentrations were determined using the enzymatic assay described above. Rates of hepatic triglyceride secretion were calculated from the time-dependent linear increases in serum triglyceride concentration.

### Statistical analysis

Data are presented as mean values with error bars representing SEM. Statistical significance was determined by using two-tailed unpaired Student's two-tailed $t$-tests when two groups were compared. Correlations were evaluated by Pearson's correlation coefficient analysis. Threshold values were determined by segmental linear regression. Multiple group comparisons were performed using one- or two-factor ANOVA, and individual comparisons were made with Tukey HSD or Bonferroni post hoc tests. Differences were considered significant for $^*p < 0.05$; $^{**}p < 0.01$; $^{***}p < 0.001$; $^{****}p < 0.0001$. (GraphPad Prism 8, GraphPad Software, La Jolla, CA, USA).

### Reporting summary

Further information on research design is available in the Nature Portfolio Reporting Summary linked to this article.

## Data availability

The Structure coordinates, and diffraction data generated in this study have been deposited in the PDB database under accession code 7U9D. The RNAseq data generated in this study have been deposited in the Geo database under accession code GSE224877. Source data are provided with this paper.

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

## Acknowledgements

The authors would like to thank Dr. Hongliang Li from the Institute of Model Animal (IMA), Wuhan University for the generous donation of the *Pctp*^flox/flox. Crystallographic data were collected at Southeast Regional Collaborative Access Team 22-ID Beamline at the Advanced Photon Source, Argonne National Laboratory and supported by the Department of Energy, Office of Science, Office of Basic Energy Sciences, under contract W-31–109-Eng-38. Finally, the authors would like to thank Pooja Srinivas, Jen Colucci, and Emma D'Agostino for assistance with editing of the manuscript. A F31 training fellowship from the National Institutes of Health National Institute of Diabetes and Digestive and Kidney Diseases (NIH/NIDDK), F31 DK 126435 and training grant T32 GM 008602 supported S.A.D during the duration of this work. This work was supported by the NIH (R01 DK 048873, DK 056626 and DK 103046 to D.E.C. and E.A.O.). M.L.C. was supported by T32 GM 008367-29 This study was supported in part by the Emory Integrated Metabolomics and Lipidomics Core, which is subsidized by the Emory University School of Medicine, and is one of the Emory Integrated Core Facilities. Additional support was provided by the Georgia Clinical and Translational Science Alliance of the NIH, Award UL1 TR 002378. The content is solely the responsibility of the authors and does not necessarily reflect the official views of the NIH.

## Author contributions

S.A.D., S.G.M., M.T., D.E.C., M.L.C., M.C.T., X.M., H.F., and E.A.O. designed research; S.A.D., S.G.M., M.T. M.L.C., X.M., T.B., M.B., K.M.S., Y.X., and A.S. performed research; M.T., and D.E.C. contributed tissue for analysis; S.A.D., S.G.M., T.B., and M.T. analyzed data; S.A.D., S.G.M., and E.A.O. wrote the paper; S.A.D., M.T., S.G.M., D.E.C., Y.X., and E.A.O. edited the manuscript.

## Competing interests

The authors declare no competing interests.
