## [Peer Review File · Nature Communications]

Ligand dependent interaction between PC-TP and PPAR δ mitigates diet-induced hepatic steatosis in male miceREVIEWER COMMENTS

Reviewer #1 (Remarks to the Author):

Deletion of phosphatidylcholine transfer protein (PC-TP), a protein involved in the transfer of PC from the ER to other cellular membranes, has been shown to improve various metabolic outcomes. Here the authors extended on previous findings by generating a liver-specific KO mouse model. KO mice showed reduced adiposity, reduced liver and muscle ectopic lipid accumulation, which the authors suggest is related to reduced PPAR δ -PC-TP binding and increased PPAR δ activity. The study is interesting and adds to a growing body of literature on the importance of hepatic lipid metabolism and PPARs in metabolic disease. While the cell biology studies are mostly sound, there are various issues with the animal experiments and the related interpretation of results, as detailed below:

- Figure 1 is low quality and the information provided within the whole can be compressed into one panel, with the remainder moved to the supps. This figure is further of low quality, with panels E, F, I, J almost impossible to read.

- In the second paragraph of the results, the authors introduce the LKO mouse model, but do not provide any information within the text on how the mouse model was generated. I notice that this is mentioned in the figure legend only. The authors used an AAV8 driven by a TBG promoter, and specify this mouse model as 'liver-specific deletion'. This is not correct. Using such an approach leads to knockdown of a target gene/protein specifically in hepatocytes, and can therefore not be termed 'liver-specific'. In addition, an AAV approach does not lead to a 100% full deletion of the protein. Where is the quantification of knockdown efficiency? Where are the gene/protein expression data for other tissues, in addition to the kidney, showing AAV and knockdown specificity?

- For Figures 1D-E, the authors should provide the actual tolerance test data. Were there differences in baseline glucose levels? Just showing AUCs does usually not justify the 'real' data. In this respect, for the AUC ITT, what was the time points used for AUC calculation? Depending on that, AUC data can have very different meanings, and the actual tolerance data are essential.

- In Figure S1, the authors assessed some lipids in liver and plasma. However, as this mouse model is likely to impact on phospholipid composition, a more detailed examination of the lipidome is warranted, including detailed changes in various phospholipid classes. In addition, Figure S1E is misleading, as the y-axis does not start at '0'. Please amend. This statement is further relevant for all subsequent HFD data, where no full profile of changes in the hepatic lipidome has been provided.

- Lines 196ff. The authors state 'The HFD L-Pctp $^{-/-}$ mice do not have major alterations in fatty acid oxidation, energy expenditure or respiration exchange ratio (Figure 4A-C), and we hypothesized that the reductions in body weight or lipid content observed in this group of mice may arise from increased energy utilization in skeletal muscle.' If there is absolutely no difference in EE between the groups, how can energy utilization be increased in muscle? Also, there is no difference in RER, so not likely that

substrate utilization is affected in these mice. As stands, this hypothesis has no ground.

- Figure 5. The authors claim that PC-TP specifically interacts with PPAR δ , however, looking at the heatmap analysis, there is similarly strong interaction with the other PPAR isoforms. Given that hepatic PPARs have overlapping transcriptional targets, how can the authors assure that the phenotypic changes are specifically due to the interaction with PPAR δ , particularly as previous studies in global PC-TP KO mice show a PPAR α phenotype? Also, the heatmap is only n=1 per observation, questioning the reproducibility of the data. Are the PC-TP metabolic effects lost in mice/cells if PPAR α is deleted, which would support the idea that it's a PPAR δ -specific interaction effect?

- Lines 204ff. In this paragraph, the authors describe changed in exercise performance in the LKO mice. However, in the next sentence they suggest that 'Given the phenotypic similarity of HFD L-Pctp $^{-/-}$ mice to activation of hepatic PPARs, we performed a PPAR qRT-PCR array on livers'. This assessment is not well placed in the context of the investigation, and there is no information provided on how the LKO mice are phenotypically similar to a hepatic PPAR phenotype.

- Figure S2E. Mistake in units for plasma insulin? Seems awfully high.

- Figure 3A. Please show the liver weight in grams (not corrected for body weight), and adjust the y-axis (starting from '0') to allow for accurate estimation of the observed changes.

- Figure 3D. It is stated in the figure that TG in muscle was assessed, but I assume this is not correct?

- Figure 4A: spelling mistake on y-axis. Also, a further spelling mistake in Figure 4 Legend: 'Promethean' should be 'Promethion'

- Figure 4C. The y-axis is incomplete. What are the units for plasma beta hydroxybutyrate?

- Abstract: '...hepatic PC-TP activity influences lipid metabolism in the muscle'. This sounds like direct impact of PC-TP on muscle. However, this is more likely related to changes in VLDL release and/or VLDL composition. Please reword.

- The rationale for this study within the introduction is not convincing. The authors state that LKO mice 'showed induction of PPAR controlled transcripts', and suggest that 'PC-TP represses PPAR δ '. Given that previous literature points to PPAR α being a primary regulator of hepatic lipid metabolism and that PC-TP has been shown to interact with PPAR α , wouldn't this be the primary hypothesis to be investigated?

- This study has a lot of self-citations, some not directly related to the respective statement. For example, in lines 113-114 the authors point to global KO mice showing improvements in metabolic health and cite 6 of their papers, including a study where PC-TP was firstly cloned and characterized, but is not relevant to the statement in the text. Excessive self-citations should be discouraged.

Reviewer #2 (Remarks to the Author):

have read the paper with enthusiasm. It reports data supporting an interesting and original mechanism whereby ligand-bound STARD2 represses the transcription activated by PPAR δ . This model sheds light on the poorly understood mechanism of ligand transport and “ligand-binding induced-signaling” (or protein-protein interactions) involving START domains. Although very appealing and logical, the mechanism would benefit from a deeper assertion, or discussion, of the exact role of ligand binding. In fact, apo-START domains populate an open state (intermediate) to allow for ligand binding and once inside the buried binding site, the ligand quenches this opening (as beautifully demonstrated by the authors) through the establishment stabilising interactions with specific side-chains. Overall the structural changes are minute (between the apo and the holo-states) making it difficult to anticipate that the interaction with PPAR δ would be modulated by the ligand or the ligand bound structure of STARD2. On the other hand, it is possible that the intermediate state (in search for its ligand i.e. PC) has a tendency to remain adsorbed at membranes. Hence, in the absence of PC and in the case of the mutants, it is (or is it) possible that apo-STARD2 stays adsorbed to the ER and is unable to translocate to the nucleus where it can repress PPAR δ . In this case, the role of the ligand would consist in allowing for the holo-state to lose its affinity for membranes, dissociate from them in order to be able to translocate to the nucleus. This is my only issue with the paper as I believe it reports technically and scientifically sound results that are highly original and important to the field of the function of START domains at large.

Reviewer #3 (Remarks to the Author):

1. The authors carefully evaluated the interaction between Pctp deletion and PPAR δ in the liver. However, it is unclear whether anti-obesity effect of hepatic Pctp deletion is actually mediated by hepatic PPAR δ activation. To validate this, in vivo experiment is essential.
2. Figure 3E: Hepatic free cholesterol is increased in KO mice. This data does not support the authors' hypothesis that Pctp KO reduced hepatic lipids and ensuing toxicity.

Reviewer #1 (Remarks to the Author):

Deletion of phosphatidylcholine transfer protein (PC-TP), a protein involved in the transfer of PC from the ER to other cellular membranes, has been shown to improve various metabolic outcomes. Here the authors extended on previous findings by generating a liver-specific KO mouse model. KO mice showed reduced adiposity, reduced liver and muscle ectopic lipid accumulation, which the authors suggest is related to reduced PPAR δ -PC-TP binding and increased PPAR δ activity. The study is interesting and adds to a growing body of literature on the importance of hepatic lipid metabolism and PPARs in metabolic disease. While the cell biology studies are mostly sound, there are various issues with the animal experiments and the related interpretation of results, as detailed below:

Response: We thank the Reviewer for their valuable insight and careful eye. We believe that incorporating the considerations brought up by this Reviewer has greatly improved the impact of the final manuscript.

- Figure 1 is low quality and the information provided within the whole can be compressed into one panel, with the remainder moved to the supps. This figure is further of low quality, with panels E, F, I, J almost impossible to read.

Response: We apologize and believe that this error occurred upon electronic submission. We have remade the figures, enlarged text, moved panels C-F to the supplement, and preserved high resolution during PDF generation.

Figure 1. RNAseq comparing *Pctp*^{-/-} and WT chow and MCD fed mice. RNA from chow and MCD fed WT (n = 3) and *Pctp*^{-/-} (n = 3) mouse livers was used for RNAseq analysis. **A & B)** Differentially expressed genes (DEGs) for chow fed and MCD fed mice respectively, comparing WT to PC-TP KO. Heat Map shows the distribution of PPAR δ controlled DEGs **C & D)** Enrichr analysis of DEGs comparing the effect of diet on each genotype compared to transcription factor CHIP-seq databases (CHEA), for WT and *PCTP*^{-/-} respectively.^{(21) (22)} **E & F)** Enrichment analysis of altered metabolic pathways (KEGG) determined by cross referencing the statistically significantly altered genes comparing the effect of diet on each genotype, for WT and *PCTP*^{-/-} respectively. Blue denotes a PPAR related process.^{(22) (23)}

Supplemental Figure 1. RNAseq comparing *Pctp*^{-/-} and WT chow and MCD fed mice. RNA from chow and MCD fed WT (n = 3) and *Pctp*^{-/-} (n = 3) mouse livers was used for RNAseq analysis. **A & B)** Enrichr analysis of DEGs from chow fed and MCD fed mice respectively, mice were compared to know transcription factor CHIP-seq databases (CHEA). **C & D)** Enrichment analysis of altered metabolic pathways (KEGG) determined by cross referencing the statistically significantly altered genes for chow and MCD fed mice respectively.

- In the second paragraph of the results, the authors introduce the LKO mouse model, but do not provide any information within the text on how the mouse model was generated. I notice that this is mentioned in the figure legend only. The authors used an AAV8 driven by a TBG promoter, and specify this mouse model as 'liver-specific deletion'. This is not correct. Using such an approach leads to knockdown of a target gene/protein specifically in hepatocytes, and can therefore not be termed 'liver-specific'. In addition, an AAV approach does not lead to a 100% full deletion of the protein. Where is the quantification of knockdown efficiency? Where are the gene/protein expression data for other tissues, in addition to the kidney, showing AAV and knockdown specificity?

Response: We thank the reviewer for correcting our terminology surrounding knockdown specificity. We have corrected this and added the following text in the main body of the manuscript and methods:

"L-Pctp^{-/-} mice were generated through i.v. injection of AAV8-TBG-Cre into Pctp^{flox/flox} mice. AAV8 harboring CRE under the TBG promoter ensures hepatocyte-specific deletion. Mice were challenged with either high fat diet (HFD) or normal diet (ND) (Figure 2A-B)."

"Tissue specific knockdown mice were created by a LoxP/Cre system C56BL6 background mice with Pctp flanked by two LoxP sites (Pctp^{flox/flox}) using a Crispr/Cas9 set up (Supplemental Figure 5). Pctp^{flox/flox} were transduced with AAV8 harboring a vector for Cre recombinase driven by the TBG promoter to generate hepatocyte specific deletion of Pctp^{-/-} (L-Pctp^{-/-}). Similarly, control mice were treated with equivalent titer of empty AAV8. These mice were viable and displayed no apparent developmental abnormalities."

Knockdown efficiency was assayed via western blot, comparing liver and kidney isolated from KO and control mice.”

Additionally, we have provided immunoblots showing efficacious knockdown in the liver, without effecting the kidney (Figure 2A). The human thyroxine binding globulin (TBG) promoter is a hepatocyte-specific promoter, which confers persistent and specific hepatic transgene expression for up to several months following integration as shown in previous literature [22820390; 20486773]. Recent work exploring hepatocellular genome editing through Cre-Lox recombination using Cre expressing AAV vectors, addressed off-target effects of these systems. The authors concluded that AAV8-TBG vectors are a reliable and efficient tool for hepatocyte-specific genetic manipulation with minimal off-target effects [PMC8487635]. Their unbiased transcriptomics analysis demonstrated a lack of major off-target effects within hepatocytes when using this vector system.

- For Figures 1D-E, the authors should provide the actual tolerance test data. Were there differences in baseline glucose levels? Just showing AUCs does usually not justify the ‘real’ data. In this respect, for the AUC ITT, what was the time points used for AUC calculation? Depending on that, AUC data can have very different meanings, and the actual tolerance data are essential.

Response: We thank the Reviewer for this thoughtful consideration and have appended these data to the supplement to allow for a more encompassing view of changes in our tolerance tests (New Supplemental figure 2 D-F). We do observe baseline differences in glucose levels between the WT and KO mice; however, these are not statistically significant.

Supplemental Figure 2. Additional characterization of the *L-Pctp*^{-/-} mouse fed HFD. **A) Growth curve of *L-Pctp*^{-/-} (KO) and WT mice on HFD throughout the period of diet intervention, with quantification of area under the curve provided in the insert (n=15, T-Test, SEM). **B & C)** Food intake and water intake of HFD cohort measured over one day show no significant difference in nutrient consumption to explain changes in weight (n=6, T-Test, SEM). **D-F)** Curves for the GTT, ITT and PTT used for determining area under the curves (n=4, SEM). **G & H)** Measure fasting glucose and insulin in KO and WT mice on HFD (n=5, T-test, SEM). **I)** qRT-PCR of cDNA generated from the livers of WT and KD mice fed HFD shows significant alteration in several genes known to be regulated by PPARs. (n=5, Two-way ANOVA, SEM).**

- In Figure S1, the authors assessed some lipids in liver and plasma. However, as this mouse model is likely to impact on phospholipid composition, a more detailed examination of the lipidome is warranted, including detailed changes in various phospholipid classes.

We previously reported an in-depth characterization of phospholipid abundance changes was reported in the whole-body PC-TP KO [29202465]. This study showed that global PC-TP deletion leads to an accumulation of PC in the endoplasmic reticulum (ER), especially PCs with polyunsaturated fatty acyl chains. While present, the effect on the glycerophospholipid composition was minor. We expect a similar alternation of hepatic glycerophospholipid content in our model of PC-TP deletion although these changes are likely attenuated as this paper utilized a whole body, developmental knock out compared to transient knock down used in our studies. In a subsequent paper, we are planning a deeper investigation of the changes in the lipidome including a search for phospholipids that may modulate the PC-TP-PPAR δ interaction.

-In addition, Figure S1E is misleading, as the y-axis does not start at '0'. Please amend. This statement is further relevant for all subsequent HFD data, where no full profile of changes in the hepatic lipidome has been provided.

Response: We thank the reviewer for catching this and have changed the axis of this figure (below).

- Lines 196ff. The authors state 'The HFD L-Pctp^{-/-} mice do not have major alterations in fatty acid oxidation, energy expenditure or respiration exchange ratio (Figure 4A-C), and we hypothesized that the reductions in body weight or lipid content observed in this group of mice may arise from increased energy utilization in skeletal muscle.' If there is absolutely no difference in EE between the groups, how can energy utilization be increased in muscle? Also, there is no difference in RER, so not likely that substrate utilization is affected in these mice. As stands, this hypothesis has no ground.

Response: While our data does not indicate a change in muscle fatty acid metabolism, we do detect decreases in lipid content in the liver, plasma, and muscle despite unchanged food intake. Total fat mass was also decreased in these mice further complicating the underlying phenotype. We agree that EE and RER are conclusive and have removed the aforementioned statement from the manuscript to not over interpret aspects of our data.

- Figure 5. The authors claim that PC-TP specifically interacts with PPAR δ , however, looking at the heatmap analysis, there is similarly strong interaction with the other PPAR isoforms.

Response: We apologize that the heat map coloring made it challenging to interpret and we have re-colored the graph on a more appropriate scale permitting better visualization of the difference between STARD2 and other PPAR isoforms (Figure 5B). Quantification of these same data in a bar graph (Figure 5C), shows a clear difference in the preference of PPAR δ , with other isoforms barely interacting with PC-TP.

Given that hepatic PPARs have overlapping transcriptional targets, how can the authors assure that the phenotypic changes are specifically due to the interaction with PPAR δ , particularly as previous studies in global PC-TP KO mice show a PPAR α phenotype? Also, the heatmap is only n=1 per observation, questioning the reproducibility of the data.

Response: These data were extracted from a high throughput screen performed with 4 technical replicates which is in-line with similar discovery-based assays. We have since incorporated a Z-score and a signal to background ratio (S/B) to allow for proper interpretation of these data. Due to the low replicate number associated with this initial screen, we have also performed follow up protein complementation assays and orthogonal bioluminescence resonance energy transfer assays, all of which confirm an interaction between PC-TP and PPAR δ (Figures 7A and 7F).

Figure 5. Discovery of a repressive PC-TP-PPAR δ complex. **A)** Depiction of the nano-PCA assay **B)** Heat map of HEK293T cells were transfected with portions of luciferase fused to LTPs or PPARs, interactions for each PPAR isoform were normalized to the highest interaction (n=1). Calculated z' and S/B noise suggest that this assay is robust using Them1-Them1, a known trimer, as a positive control and beta-catenin with FABP3 as a negative control (z'=0.795 S/B=7.03) **C)** Bar graph showing the quantification from the nano-PCA screen of known interactors for PC-TP and PPAR δ . PPAR δ (blue) interacts with PC-TP (orange) stronger than known partners (#) of either protein (SEM). **D)** To evaluate the effect of PC-TP on the activity of PPAR δ , siRNA knockdown of PC-TP was performed in Huh7 cells using targeted and scrambled siRNA. Knockdown (KD) efficiency was confirmed by Western blot. Target mRNA expression levels at 48H post KD was quantified by qRT-PCR and showed increased PPAR δ transactivation when PC-TP is transiently KD. (n=6, One way ANOVA, SEM) **E)** Luciferase reporter assay in conjunction with siRNA of PC-TP's confirms PC-TP negatively regulates PPAR δ 's transactivation (n=3, Two-way ANOVA, SEM)

Are the PC-TP metabolic effects lost in mice/cells if PPAR α is deleted, which would support the idea that it's a PPAR δ -specific interaction effect?

Response: Previous experiments have suggested an indirect effect of PC-TP deletion on other nuclear receptors such as HNF4 and PPAR α . While we agree that crossing our *L-Pctp*^{-/-} mouse with PPAR specific KO would be informative as to the role of indirect PPAR α modulation compared to direct PPAR δ modulation, we feel these studies are beyond the scope of this manuscript. Certainly, the observed phenotypes may represent perturbations in several signaling and cellular processes and we elected to pursue one of these centered on PC-TP modulating PPAR activity. We only conclude a selective, repressive interaction through our *in vitro* characterization. Finally, luciferase reporter experiments require PPAR δ overexpression which points PPAR δ dependent effect on transactivation on luciferase levels under control of a PPRE when PC-TP is overexpressed.

- Lines 204ff. In this paragraph, the authors describe changed in exercise performance in the LKO mice.

However, in the next sentence they suggest that ‘Given the phenotypic similarity of HFD L-Pctp^{-/-} mice to activation of hepatic PPARs, we performed a PPAR qRT-PCR array on livers’. This assessment is not well placed in the context of the investigation, and there is no information provided on how the LKO mice are phenotypically similar to a hepatic PPAR phenotype.

Response: We thank the reviewer for bringing this to our attention. We have added a description of the phenotypic similarities between our model and hepatic PPAR modulation prior to the aforementioned statement and believe this has improved the impact and interpretation of our study:

“Taken together our data suggest a role for PC-TP in regulating PPARs as our L-Pctp^{-/-} mouse on HFD displayed an opposing phenotype to hepatic deletion of PPARs. Deletion of hepatic PPAR α resulted in decreased beta oxidation, insulin resistance and increased liver lipid accumulation. Similarly liver specific PPAR δ KO presented with insulin resistance, dyslipidemia and steatosis.^(24, 25) While seemingly redundant, PPAR α is thought to control these processes in the fasted state, with PPAR δ playing a more important role in the fed state.⁽²⁶⁾ To test this, we performed a PPAR qRT-PCR array on livers from HFD animals which shows an increase in PPAR regulated transcripts in L-Pctp^{-/-} mice (Figure 4J) (Supplemental Figure 2F).”

- Figure S2E. Mistake in units for plasma insulin? Seems awfully high.

Response: We thank the reviewer for catching this error which has been corrected.

- Figure 3A. Please show the liver weight in grams (not corrected for body weight), and adjust the y-axis (starting from '0') to allow for accurate estimation of the observed changes.

Response: We have changed this figure to be more in line with the Reviewer suggestion and added the normal diet (ND) as a point of comparison.

- Figure 3D. It is stated in the figure that TG in muscle was assessed, but I assume this is not correct?
- Figure 4A: spelling mistake on y-axis. Also, a further spelling mistake in Figure 4 Legend: 'Promethean' should be 'Promethion'
- Figure 4C. The y-axis is incomplete. What is the units for plasma beta hydroxybutyrate?

Response: We have addressed/ corrected these concerns and thank the reviewer for bringing them to our attention

- Abstract: '...hepatic PC-TP activity influences lipid metabolism in the muscle'. This sounds like direct impact of PC-TP on muscle. However, this is more likely related to changes in VLDL release and/or VLDL composition. Please reword.

Response: While we agree with this reviewer that this could be in part through altered composition of lipoprotein particles, we see a global loss of TG, and FFA species that are not accounted for by altered location via lipoprotein transport. We have changed the above statement to avoid over interpreting our own data focusing on a liver centric role in this process to be more in line with reviewer suggestion.

"Hepatic deletion of PC-TP reduced total adipose tissue mass with decreases in levels of triglycerides and phospholipids in skeletal muscle, liver and plasma suggesting that hepatic PC-TP activity influences whole body lipid homeostasis."

The rationale for this study within the introduction is not convincing. The authors state that LKO mice 'showed induction of PPAR controlled transcripts' and suggest that 'PC-TP represses PPAR δ '. Given that previous literature points to PPAR α being a primary regulator of hepatic lipid metabolism and that PC-TP has been shown to interact with PPAR α , wouldn't this be the primary hypothesis to be investigated?

Response: We wish to clarify a misconception relating to prior research. While PC-TP has been implicated in modulating PPAR α activity, this is an indirect effect thought to occur through altering ligand availability [20045742]. No evidence for direct binding has been shown to our knowledge. Furthermore CHIP-seq experiments support a role for PPAR α in modulating the liver in a fasted state where as hepatic PPAR δ is more important for transcriptional control of the fed state.^[26] We believe that PC-TP modulation could play an important role in this switch, as under serum starved conditions we see an enhanced interaction which should further represses PPAR δ activity. Furthermore, in our screen

there is only a weak interaction between PPAR α and PC-TP. We do, however, capture an interaction between SATRD10 and PPAR α which is supported by literature and merits further investigation.

- This study has a lot of self-citations, some not directly related to the respective statement. For example, in lines 113-114 the authors point to global KO mice showing improvements in metabolic health and cite 6 of their papers, including a study where PC-TP was firstly cloned and characterized, but is not relevant to the statement in the text. Excessive self-citations should be discouraged.

Review: We have revised the manuscript and removed superfluous citations and did not intend to neglect other researchers on this topic. We have also attempted to diversify our citations when possible and thank this reviewer for bringing this to our attention.

Reviewer #2 (Remarks to the Author):

have read the paper with enthusiasm. It reports data supporting an interesting and original mechanism whereby ligand-bound STARD2 represses the transcription activated by PPAR δ . This model sheds light on the poorly understood mechanism of ligand transport and “ligand-binding induced-signaling” (or protein-protein interactions) involving START domains. Although very appealing and logical, the mechanism would benefit from a deeper assertion, or discussion, of the exact role of ligand binding. In fact, apo-START domains populate an open state (intermediate) to allow for ligand binding and once inside the buried binding site, the ligand quenches this opening (as beautifully demonstrated by the authors) through the establishment stabilizing interactions with specific side-chains. Overall the structural changes are minute (between the apo and the holo-states) making it difficult to anticipate that the interaction with PPAR δ would be modulated by the ligand or the ligand bound structure of STARD2. On the other hand, it is possible that the intermediate state (in search for its ligand i.e. PC) has a tendency to remain adsorbed at membranes. Hence, in the absence of PC and in the case of the mutants, it is (or is it) possible that apo-STARD2 stays adsorbed to the ER and is unable to translocate to the nucleus where it can repress PPAR δ . In this case, the role of the ligand would consist in allowing for the holo-state to lose its affinity for membranes, dissociate from them in order to be able to translocate to the nucleus. This is my only issue with the paper as I believe it reports technically and scientifically sound results that are highly original and important to the field of the function of START domains at large.

Response: We thank the Reviewer for their valuable insight and agree that the consideration as to the role in apo/holo PC-TP is important. Our efforts to crystallize apo-PC-TP were not successful and the protein appears very unstable when we strip co-purified phospholipids after bacterial expression. This also hampers our ability to characterize the apo-PC-TP – PPAR δ interaction *in vitro*. Interestingly, crystallizing protein with co-purified lipids from expression results in a PC-TP-PC complex. We suspect PC-TP is capturing PC from yeast extract since *E. coli* do not synthesize this lipid. We have conducted molecular dynamics simulations of apo vs holo-PC-TP, however, the major differences in conformational fluctuations were the in the C-terminal helix. This helix is also involved in membrane association and in PC binding/transfer and truncation of these residues ablates PC transfer. We show that mutations designed to interfere with ligand binding to PC-TP reduces its ability to interact with and modulate PPAR δ , leading us to believe that this is as a result of Holo-PC-TP. Future studies are underway to investigate the interaction interface as well as the ligand dependence of the interaction.

Reviewer #3 (Remarks to the Author):

1. The authors carefully evaluated the interaction between Pctp deletion and PPAR δ in the liver. However, it is unclear whether anti-obesity effect of hepatic Pctp deletion is actually mediated by hepatic PPAR δ activation. To validate this, *in vivo* experiment is essential.

Response: While we agree with the Reviewer that this exciting study merits follow up, we do not conclude that there is an interaction between PC-TP and PPAR δ from our *in vivo* work, but merely demonstrate through multiple models that there are similarities on a phenotypic level between our model and the effects of modulating hepatic PPARs. It is through our *in vitro* work that we derive an isoform specific, repressive interaction between PC-TP and PPAR δ which could explain a part of our observed phenotype. This also doesn't negate other mechanisms of regulation that could occur through interactions between PC-TP and other proteins or indirect modulation through regulating distribution of PC which can affect membrane fluidity and receptor function. We have attenuated our language surrounding this topic as to not overinterpret our data.

2. Figure 3E: Hepatic free cholesterol is increased in KO mice. This data does not support the authors' hypothesis that Pctp KO reduced hepatic lipids and ensuing toxicity.

Response: While we see a slight increase in the hepatic cholesterol content, we also see decreases in TG and other lipid species which could explain the stark phenotype associated with KO on hepatic lipid droplet accumulation under high fat diet. We have clarified our language regarding this observation and our hypothesis:

"In addition to a reduction in fat mass and total body weight, we also observed a significant decrease in liver weight for L-Pctp^{-/-} mice fed HFD compared to WT HFD mice (Figure 3A). Under HFD, deletion of hepatic PC-TP normalized liver size to a weight comparable to WT mice fed ND. In line with this, we also observed reduced lipid droplet formation in L-Pctp^{-/-} mice fed HFD relative to WT HFD mice (Figure 3B-C), accompanied by a decrease in hepatic triglyceride (TG) content (Figure 3D-E). Taken together, this data points to a role for PC-TP in the progression of liver lipotoxicity driven by a state of overnutrition."

REVIEWER COMMENTS

Reviewer #1 (Remarks to the Author):

The authors put extensive effort into their rebuttal and the addition of new data and text. I agree with most of the changes, however, there are still various weaknesses primarily surrounding (1) off-target effects, (2) the GTT and ITT data, and (3) the hepatic lipidome, that are important but have not been addressed:

- The authors mention that baseline glucose levels were not significantly different between groups but Figure S3G contradicts that statement, showing a significant reduction in fasting blood glucose. Also, it still remains unclear what time points were utilized for the assessment of AUC during both GTT and ITT. In respect to the GTT, was a tAUC or iAUC utilized? If tAUC, the reduction in fasting blood glucose fully explains the differences between groups, meaning that mice show improved hyperglycaemia and not glucose tolerance.

For the assessment of insulin tolerance, what really matters is the drop in blood glucose within the first 30min. All further changes are secondary to the direct insulin response. In this respect, there is absolutely no difference in the ITT in the ND-fed mice, while the authors point to a significance increase. As is, the meaningful data (GTT, ITT, FBP, insulin) are hidden away in the supplement while the data with 'selective' interpretation are shown. This should be swapped to allow the reader to make up their own mind.

- This manuscript centres around understanding lipid metabolism in the liver with the authors modulating a PL-metabolism related protein. It is surprising that the authors, despite a recommendation, did not assess the hepatic lipidome (having the methodological expertise as shown through previous studies). Instead, the authors point to a global KO mouse model. While this is relevant, a global KO model would impact the lipidome across many tissues as well as in the circulation, and might not lead to the 'same' hepatic lipidome changes as in the hepatocyte-specific deletion model.

- The authors did not follow up assessing off-target effects in other tissues than the kidney but instead reference a study (on a different target protein) where off-target effects were assessed. While I understand that the AAV8-TBG is relatively specific, having worked with the virus for many years, assessment of tissue-specific overexpression/knockdown is good scientific practice.

Reviewer #2 (Remarks to the Author):

My comments have been addressed.

Reviewer #3 (Remarks to the Author):

The authors responded to reviewers' queries.

Reviewer #1 (Remarks to the Author):

The authors put extensive effort into their rebuttal and the addition of new data and text. I agree with most of the changes, however, there are still various weaknesses primarily surrounding (1) off-target effects, (2) the GTT and ITT data, and (3) the hepatic lipidome, that are important but have not been addressed:

We thank this Reviewer for their thoughtful comments and believe incorporating these changes has greatly elevated the quality of our manuscript.

- The authors mention that baseline glucose levels were not significantly different between groups but Figure S3G contradicts that statement, showing a significant reduction in fasting blood glucose. Also, it still remains unclear what time points were utilized for the assessment of AUC during both GTT and ITT. In respect to the GTT, was a tAUC or iAUC utilized? If tAUC, the reduction in fasting blood glucose fully explains the differences between groups, meaning that mice show improved hyperglycaemia and not glucose tolerance. For the assessment of insulin tolerance, what really matters is the drop in blood glucose within the first 30min. All further changes are secondary to the direct insulin response. In this respect, there is absolutely no difference in the ITT in the ND-fed mice, while the authors point to a significance increase.

As is, the meaningful data (GTT, ITT, FBP, insulin) are hidden away in the supplement while the data with 'selective' interpretation are shown. This should be swapped to allow the reader to make up their own mind.

All tolerance data was collected after the 12-week dietary intervention, this information has been added to the figure legends and the methods section, appended below.

For quantification of these data, tAUC was used. We have clarified our language surrounding this analysis, only stating improved glycemic control and not improved tolerance. We have also moved all tolerance test traces into the main body of the manuscript as per this Reviewers suggestion.

Figure 2. Characterization of the L-*Pctp*^{-/-} mouse. **A)** L-*Pctp*^{-/-} mice were generated through i.v. injection of AAV8-TBG-Cre into *Pctp*^{fllox/fllox} mice. AAV8 harboring CRE under the TBG promoter ensures hepatocyte-specific deletion. Control mice (WT) were *Pctp*^{fllox/fllox} mice treated with AAV8-TBG-LacZ through i.v. injection. Following AAV8 injection, mice were fed either ND or HFD for an additional 12 weeks prior to all analysis. Representative western blot showing hepatocyte-specific knockdown of PC-TP in L-*Pctp*^{-/-} group). **B)** Echo-RI of L-*Pctp*^{-/-} and WT mice on ND or HFD (n=6, Two-way ANOVA, SEM). **C-E)** Curves for the GTT, ITT and PTT assessed at week 12 used for mapping total area under the curves (n=4, SEM). **F)** Total area under the curve for glucose tolerance test (n=4, T-test, SEM). **G)** Total area under the curve for insulin tolerance test (n=4, T-test, SEM). **H)** Total area under the curve for pyruvate tolerance test (N=4, T-test, SEM) *, p<0.05; **, p<0.01; *** p<0.001; ****p<0.0001.

- This manuscript centres around understanding lipid metabolism in the liver with the authors modulating a PL-metabolism related protein. It is surprising that the authors, despite a recommendation, did not assess the hepatic lipidome (having the methodological expertise as shown through previous studies). Instead, the authors point to a global KO mouse model. While this is relevant, a global KO model would impact the lipidome across many tissues as well as in the circulation, and might not lead to the 'same' hepatic lipidome changes as in the hepatocyte-specific deletion model.

We agree with this Reviewer and have now performed high resolution untargeted lipidomics on livers from the same hepatocyte-specific PC-TP knockdown mouse (L-*Pctp*^{-/-}) (Supplemental Figure 4 and Supplemental Table 1). We restricted our analysis to lipids with high-confidence MS/MS based annotations and observed significant changes in only in triglyceride (TG) species complementing our bulk TG analysis (Figure 3). Most of the statistically significantly altered TGs contain medium to long acyl-chains, with desaturation that may indicate alteration in essential fatty acid homeostasis, although more targeted methods are required to confirm this. We see no changes in the abundance or composition of glycerophospholipids in line with prior studies suggesting that PC-TP affects ER to plasma membrane lipid distribution rather than global glycerophospholipid levels.

- The authors did not follow up assessing off-target effects in other tissues than the kidney but instead reference a study (on a different target protein) where off-target effects were assessed. While I understand that the AAV8-TBG is relatively specific, having worked with the virus for many years, assessment of tissue-specific overexpression/knockdown is good scientific practice.

We have included additional tissues such as heart muscle and brown adipose tissue in our western blot analysis and confirm the specificity of the knockdown. These data are now included in figure 2 (appended above).

Reviewer #2 (Remarks to the Author):

My comments have been addressed.

Reviewer #3 (Remarks to the Author):

The authors responded to reviewers' queries.

REVIEWERS' COMMENTS

Reviewer #1 (Remarks to the Author):

I congratulate the authors on this high-impact study. All my comments have been adequately addressed, and I have no further suggestions.

REVIEWERS' COMMENTS

Reviewer #1 (Remarks to the Author):

I congratulate the authors on this high-impact study. All my comments have been adequately addressed, and I have no further suggestions.

We thank this reviewer so much for the valuable time they spent to reviewing this manuscript.